



# Estimating immediate post-fire carbon fluxes using the eddy-covariance technique

Bruna R. F. Oliveira[1], Carsten Schaller[2], J. Jacob Keizer[1], Thomas Foken[3]

[1] Earth surface processes team, Center for Environmental and Marine Studies (CESAM), Department of Environment and Planning, University of Aveiro, Aveiro, Portugal
[2] Climatology Research Group, Institute of Landscape Ecology, University of Münster, Münster, Germany
[3] Bayreuth Center of Ecology and Environmental Research (BayCEER), University of Bayreuth, Bayreuth, Germany

*Correspondence to*: Thomas Foken (thomas.foken@uni-bayreuth.de)

**Abstract.** Wildfires typically affect multiple forest ecosystem services, with carbon sequestration being affected both directly, through the combustion of vegetation, litter and soil organic matter, and indirectly, through perturbation of the energy and matter balances. Post-fire carbon fluxes continue poorly studied at the ecosystem scale, especially during the initial window-of-disturbance when changes in environmental conditions can be very pronounced due to the deposition and subsequent mobilization of a wildfire ash layer and the recovery of the vegetation. Therefore, an eddy-covariance system was installed in a burnt area as soon as possible after a wildfire that had occurred on 13 August 2017, and has been operating from the 43rd post-fire day onwards. The study site was specifically selected in a Mediterranean woodland area dominated by Maritime Pine stands with a low stature that had burnt at high severity.

The carbon fluxes recorded during the first post-fire hydrological year tended to be very low, so that a specific procedure for the analysis and, in particular, gap filling of the eddy covariance data had to be developed. Still, the carbon fluxes varied noticeably during the first post-fire year, broadly revealing five consecutive periods. During the rainless period after the wildfire, fluxes were reduced but, somewhat surprisingly, indicated a net assimilation. With the onset of the autumn rainfall, fluxes increased and corresponded to a net emission, while they became insignificant with the start of the winter. From the mid winter onwards, net fluxes became negative, indicating a weak carbon update during spring followed by a strong uptake during summer. Over the first post-fire year as a whole, the cumulative net ecosystem exchange was -347 g C m$^{-2}$, revealing a relatively fast recovery of the carbon sink function of the ecosystem. This recovery was mainly due to understory species, both resprouter and seeder species, since pine recruitment was reduced.

Specific periods during the first post-fire year were analyzed in detail for improving process understanding. Perhaps most surprisingly, dew formation and, more specifically, its subsequent evaporation was found to play a role in carbon emissions during the rainless period immediately after fire, involving a mechanism distinct from de-gassing of the ash/soil pores by infiltrating water. The use of a special wavelet technique was fundamental for this inference.



# 1 Introduction

The increasing frequency and intensity of extreme climate events (IPCC, 2018) is contributing to an increase in frequency and severity of wildfires (Flannigan et al., 2013; Keeley and Syphard, 2016). Such unprecedented wildfire regimes have been causing widespread concerns about their socio-economic and environmental impacts, including damages to ecosystems and

the services they provide (Moritz et al., 2014). An important ecosystem service that is impacted by wildfires is carbon sequestration by forests (Campbell et al., 2007; Restaino and Peterson, 2013). Thereby, wildfires can interfere with forest policy and management goals for climate change mitigation (Restaino and Peterson, 2013; Ruiz-Peinado et al., 2017).

Wildfires impact forest carbon pools not only directly through combustion of vegetation and litter biomass and soil organic matter, but also indirectly through disturbance of energy, water and carbon fluxes (Sommers et al., 2014; Stevens-Rumann et

al., 2017). These indirect effects are particularly difficult to assess as they depend on a complexity of factors related to fire severity, forest type, post-fire land management and post-fire environmental conditions (De la Rosa et al., 2012; Santana et al., 2016; Serrano-Ortiz et al., 2011). Furthermore, these effects can be long lasting, as well illustrated by Dore et al. (2008), finding that a 10-year old burnt site was still a carbon source.

In their review of 2013, Restaino and Peterson (2013) argued that relatively few studies had assessed post-fire carbon dynamics

through the measurement of carbon fluxes as opposed to changes in carbon pools, and that relatively few of these flux studies had used the eddy-covariance (EC) technique. Marañón-Jiménez et al. (2011) likewise affirmed that post-fire studies of soil carbon effluxes were relatively abundant. To date, EC studies following wildfires continue to be scarce (Amiro, 2001; Dadi et al., 2015; Dore et al., 2008; Mkhabela et al. 2009; Serrano-Ortiz et al., 2010; Sun et al., 2016). Furthermore, only the study of Sun et al. (2016) concerned the immediate post-fire period, with EC measurements starting from the 4th month after fire. To

address this knowledge gap, this study aimed to instrument a Maritime Pine forest with a flux tower as soon as possible after a wildfire, in particular a high-severity wildfire as indicated by complete consumption of the pine crowns (following Maia et al., 2012). Because of the lack of comparable studies and of the marked changes that were expected in both abiotic and biotic conditions due to mobilization of wildfire ash and/or vegetation recovery, a specific objective of the present study was to carry out an in-depth analysis of the obtained EC data.

## 2 Materials and Methods

### 2.1 Study area

The study area (N39º 37' W08º 06') was located 8 km to the southwest of the geodetic center of Portugal, in a Mediterranean climate zone at the transition of Köppen-Geiger classes Csa and Csb, with dry summers and an average temperature of 22ºC in the warmest month (Kottek et al., 2006). The study area was selected on 2 September 2017 for three main reasons: (i) having

been severely affected by a recent wildfire; (ii) being dominated by Maritime Pine (*Pinus pinaster* Ait.) stands of comparatively low stature (≤10 m); (iii) consisting of relatively flat terrain within the presumed footprint area. Tree species and height were

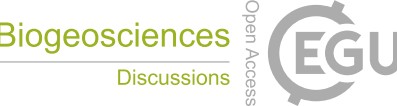

preselected based on the available, slim tower of 12 m high. The study area was a plateau of sedimentary sandstone deposits, at an elevation of 240-250 m a.s.l. (Supplementary Figure S1-a), with slopes of up to 5° over an extension of approximately 10 ha (Supplementary Figure S1-b).

The wildfire affecting the study area occurred on 13 August 2017 and burnt some 12.5 km$^2$ of woodland in total (ICNF, 2017). According to the European Forest Fire Information System (EFFIS, 2017), the fire severity in the study area varied between moderate and high. Fire severity was also assessed in the field, on 9 September 2017, along a 500 m transect that was laid out to the west of the slim tower, in the central part of the presumed footprint area. More specifically, severity was determined at 5 points along the transect and, at each transect point, for three plots centered on the nearest pine tree and the nearest shrub,

and the inter-patch in between. At all 5 transect points, crown consumption of the pine trees exceeded 75%, undergrowth vegetation and litter were fully consumed, and wildfire ash was predominantly black. The ash layer varied in depth between 0.4 and 1.0 cm, and in cover between 55 and 100%. Soil burn severity at the 15 plots was classified according to Vega et al. (2013), and ranged from moderate-to-high (class 3) at 8 plots to high (class 4) at 7 plots.

A map of tree species in the study area was made through photo-interpretation of an ortho-photomap produced from aerial

photographs that had been acquired with a RGB camera mounted on a drone (DJI Phantom 3) on 18 July 2018. Maritime Pine stands covered 90 % of the presumed footprint area, while Eucalypt (*Eucalyptus globulus*) stands occupied the remaining 10 % (Figure 1). Also during July 2018, the pine stands in the foot print area were characterized, using 5 plots of 5 m x 5 m centered on the pine trees of the above-mentioned fire severity assessment. Median height and diameter-at-breast-height of the burnt pine trees ranged from 4.6 to 6.7 m and from 2.5 to 4.3 cm, respectively. The densities of living pines varied from 0.24

to 1.72 trees m$^{-2}$ before fire to 0.12 to 1.04 seedlings m$^{-2}$ after fire. This decrease in density by the fire could be explained by the young age of the stands (in median, 12-15 years), in combination with fire damage to the (aerial) seedbank, in line with the extensive combustion of the pine crowns (Maia et al., 2012). The density of resprouting shrubs ranged from 0.0 to 0.16 shrubs m$^{-2}$ (details on vegetation composition are given in Supplement Table S1.

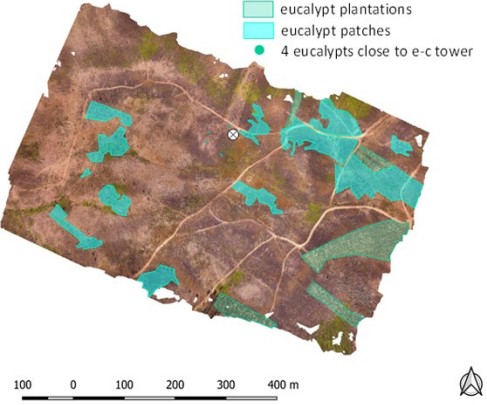

**Figure 1. Ortho-photomap of the study area showing the Eucalypt stands as well as the 4 individual Eucalypt specimens close to the slim tower (cycle with cross), the rest of the area consisting of Maritime Pine stands**



## 2.2 Experimental set-up

After obtaining authorization of the land owners, the study area was instrumented with an eddy covariance system mounted
on a slim tower and powered by 4 solar panels. The system was installed on 22 September 2017 and started operating four
days later, i.e. 43 days after the wildfire. The exact location of the tower and the height and orientation of the gas analyzer and
3D anenometer were determined on the basis of the available regional climate information, indicating a prevalence of NW
winds. This was confmed by the measurements during the first post-fire year, as shown in Figure 2.

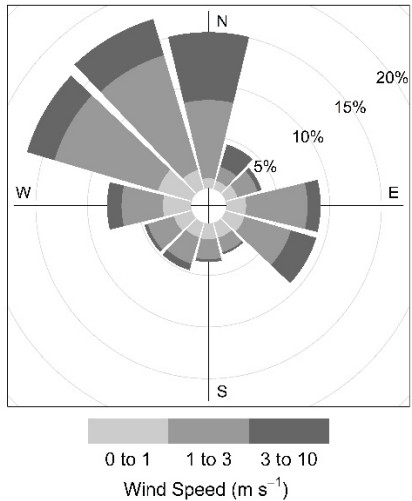

**Figure 2. Wind rose of the study area over the first post-fire hydrological year (01 October 2017 – 30 September 2018), based on the
sonic anemometer measurements at 11.6 m height.**

A picture of the tower immediately after installation is shown in Figure 3, while the installed devices are listed in Table 1. The
– standing - pine trunks in the immediate surroundings of the tower were approximately 8 m high, and this was used as
"canopy" height in all calculations, together with a zero-plane displacement of 3.8 m. The data for the calculation of the
turbulent fluxes were sampled and stored at 20 Hz using a CR6 data logger from Campbell Sci. Ltd., while the fluxes were
calculated over 30-minute intervals. All other data were sampled at 0.02 Hz, stored at 15-minute intervals and then averaged
over the 30-minutes intervals, except for rainfall. Rainfall was recorded using two automatic rainfall gauges, with a 0.2 mm
resolution and then summed over the 30-minute intervals. In addition to the soil moisture/temperature station immediately next
to the tower listed in Table 1, five soil moisture/temperature stations were installed along the above-mentioned transect. Each
station comprised 4 EC5 soil moisture sensors and 1 GS3 soil moisture/temperature sensor that were connected to an Em50
data logger, all DECAGON Devices. In this study, only the data from five of the soil moisture sensors were used, i.e. those
installed at 2.5 cm depth at the inter-patches nearest to the transect points.





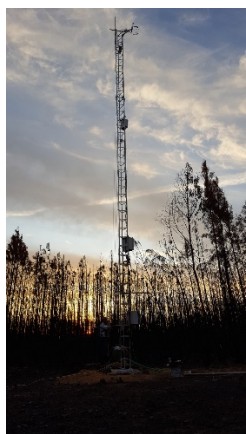

**Figure 3. The 12 m slim tower with the eddy-covariance system immediately after its installation (Photograph: J. Jacob Keizer, 22 September 2017)**

**Table 1. Meteorological sensors mounted on the slim tower and in its immediate surroundings**

| Height | Parameter | Sensor | Manufacturer | Remarks |
|---|---|---|---|---|
| 11.8 m | Wind vector | Sonic anemometer | Campbell Sci Inc. | 20 Hz data |
| | Sonic temperature | CSAT3 | | |
| 11.8 m | Water vapour and carbon dioxide | LI 7500A | LiCor Biosciences | 20 Hz data |
| 1 m | Net radiation | NR lite-2 | Kipp & Zonen | 20 Hz data |
| 2 m | Temperature and Relative Humidity | HMP45 | Vaisala Oyj | 0.02 Hz data |
| -2.5; -7.5; -10; -20; -30 cm | Soil temperature and volumetric water content; | GS3 sensor linked to a Em50 data logger | DECAGON Devices | 1.5 m from tower; 0.02 Hz data |
| 20 cm | Rain gauge 1 and Rain gauge 2 | Tipping-bucket rain gauge with 0.2 mm resolution connected to HOBO event data logger | Pronamic (rain gauge) and Onset (data logger) | 1 km to the W of tower |

## 2.3 Data calculation and quality control

This study was limited to the first hydrological year after the wildfire, from 01 October 2017 to 30 September 2018. The preceding data from 26 to 30 September 2017 were only used for one of the specific cases that were analyzed in more detail to improve process understanding (Section 3.3).



### 2.3.1 Eddy covariance data

The eddy covariance (EC) method is well-established to calculate energy and matter fluxes between the atmosphere and the
underlying surface (Aubinet et al., 2012). Therefore, the applied procedures are only briefly described. The 30-minute EC
values were calculated automatically by the Campbell EasyFlux software in the CR6 data logger but just for checking the
operational status of the system. The calculations presented here were done using the software package TK3 (Mauder and
Foken, 2015), which was found to compare well with other packages (Fratini and Mauder, 2014; Mauder et al., 2013). All
corrections to the EC values were done following the recommendations by Foken et al. (2012), and involved spike detection,
time delay correction, double rotation, and SND- and WPL-correction (Schotanus et al., 1983; Webb et al., 1980). The quality
of the flux data was checked following the method by Foken and Wichura (1996) and using the latest published version of the
flagging system (Foken et al., 2012). This procedure was also used in gap filling (Ruppert et al., 2006). Finally, all 30-minute
values were checked by means of an MAD analysis (Papale et al., 2006).

The footprint area was determined with the model of Kormann and Meixner (2001). More than 25 % of the EC measurements
coincided with more than 80 % of the Maritime Pine stands, whereas 60 % of the measurements coincided with more than
60% of the Maritime Pine stands (Figure 4). Both percentages are high in comparison with literature (Göckede et al., 2008).

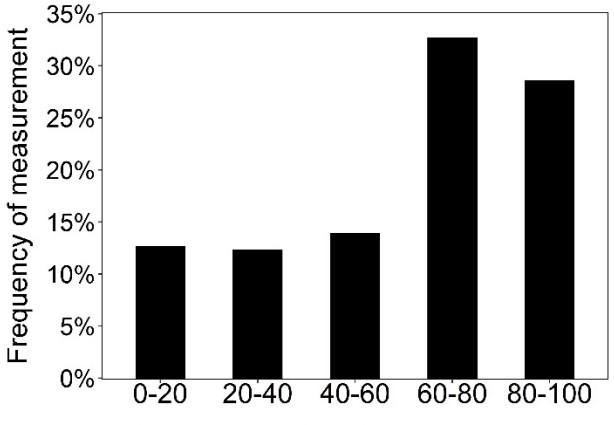

**Figure 4. Distribution of the eddy covariance measurements during the first post-fire hydrological year over five classes of footprint
area based on the degree of correspondence to the Maritime Pine stands**


Basic EC data analysis and, in particular, gap filling was done using the data that met quality classes 1-6 (Foken et al., 2012)
and had footprint areas that consisted for more than 80 % of the Maritime Pine stands, as shown in the flow diagram of
Supplementary Figure S7. The same criteria were used for selecting the specific cases presented in section 3.3. For the
cumulative fluxes over the first post-fire year, all EC data with quality classes 1-8 were combined with gap-filled data,
following the procedure shown in Supplementary Figure S8.





### 2.3.2 Gap-filling of respiration and assimilation

The gap-filling procedure used for substituting missing data as well as data of low quality was based on the Lloyd-Taylor and Michaelis-Menten functions, as it is a well-established procedure (Falge et al., 2001; Gu et al., 2005; Hui et al., 2004; Lasslop et al., 2010; Moffat et al., 2007; Reichstein et al., 2005).

The Lloyd-Taylor function was used to calculate respiration, $Q_R$:

$$Q_R = Q_{R,10} \exp\left[E_0 \left(\frac{1}{283.15-T_0} - \frac{1}{T-T_0}\right)\right] \qquad (1)$$

Where $T$ is the temperature, $Q_{R,10}$ is the respiration at 10 °C, $T_0 = 227.13$ K and describes the temperature dependence of respiration (Falge et al., 2001; Lloyd and Taylor, 1994). The parametrization of $Q_{R,10}$ and $E_0$ was done using the nighttime $CO_2$ flux data, when assimilation is zero. The nighttime period is typically determined based on a threshold of global radiation but,

since only net radiation was measured in this study, nighttime was defined here as the time window from 10 pm to 04 am UTC (UTC being the nearly local time at the study site). The parameter values were then determined using the median fluxes of 5K temperature intervals.

The parametrization of the carbon uptake at daytime, $Q_{c, day}$, was done with the Michaelis-Menten function (Falge et al., 2001; Michaelis and Menten, 1913), which must be determined for separate classes of temperature and global radiation:

$$Q_{c,day} = \frac{a\,R_n\,Q_{c,sat}}{a\,R_n + Q_{c,sat}} + Q_{R,day} \qquad (2)$$

where $Q_{c,sat}$ is the carbon flux at light saturation, $R_n$ is the net radiation corrected with the longwave net radiation (as global radiation was not measured), as detailed in. 3.2.2, $Q_{R, day}$ is the respiration at daytime, and $a$ is the linear slope of the assimilation function beginning at a global radiation of 0 W m$^{-2}$ (Falge et al., 2001; Michaelis and Menten, 1913). The constants $a$ and $Q_{c,sat}$ were determined by multiple regression, for separate classes of temperature and corrected net radiation.

### 2.3.3 Turbulent fluxes with high temporal resolution

The wavelet-based flux computation method was used to analyze a short-term flux event with non-steady-state fluxes during the rainless period immediately after the wildfire. This method offers the possibility to determine fluxes with a temporal resolution as high as 1 minute (Schaller et al., 2017). The wavelet method agrees well with the EC method for steady-state conditions, and was successfully applied to analyze short events of high methane fluxes in the recent studies of Göckede et al.

(2019) and Schaller et al. (2019).

The wavelet method in this study applied the Mexican-hat wavelet, as it provides an excellent resolution of the fluxes in the time domain and identifies the exact moment in time when single events occur (Collineau and Brunet, 1993). The wavelet method was applied using spike-free and non-rotated raw data and the assumption of a negligible mean vertical wind velocity was fulfilled. Furthermore, the cone of influence (Torrence and Compo, 1998) was estimated to guarantee that the results were

not affected by edge effects.




### 2.3.4 Ground heat flux

The ground heat flux was calculated from the above-mentioned soil temperature measurements (Table 1) and the heat storage of the topsoil, from the soil surface to a depth of 15 cm (Liebethal and Foken, 2007; Yang and Wang, 2008):

$$Q_G(0) = -\lambda \frac{\partial T}{\partial z}\Big|_{z=-0.15} + \int_{-0.15}^{0} c_v(z)\, T_s(z)\, dz \qquad (3)$$

where $T_s$ is the soil temperature, $z$ is the depth, $\lambda$ is the thermal molecular conductivity of the soil, and $c_v$ is the soil's volumetric heat capacity. The accuracy of this method is comparable to that using heat flux plates (Liebethal et al., 2005). The soil temperature at 15 cm depth was calculated as the average of the temperatures at 10 and 20 cm depth, while the thermal conductivity was estimated as the mean value at the same depth, using the temperature dependent data given by Hillel (1998). The heat capacity was computed using the equation proposed by de Vries (1963), ignoring the organic soil component:

$$c_v = c_{v,m} x_m + c_{v,w}\theta \qquad (4)$$

where $\theta$ is the volumetric soil water content,, $c_{v,m}$ and $c_{v,w}$ are the heat capacities of the mineral soil compounds ($1.9 \cdot 10^6$ J m$^{-3}$ K$^{-1}$) and soil water ($4.0 \cdot 10^6$ J m$^{-3}$ K$^{-1}$), respectively, and $x_m$ is the bulk density of the mineral compounds (0.566 m$^3$ m$^{-3}$), which was estimated from dry bulk density measurements of the soil and an assumed particle density of the mineral soil of 2650 kg m$^{-3}$.

### 2.3.5 Volumetric soil water content classes

The 30-minutes values of volumetric soil water content (VWC) of each of the 5 inter-patch sensors along the transect were first rescaled to a zero-minimum value. This was done by summing the negative minimum value over the first post-fire hydrological year (ranging from -0.07 to -0.01 m$^3$ m$^{-3}$) or, in one case, subtracting the positive minimum value (0.01). The median of the rescaled values of the 5 sensors was then calculated for each timestamp (Supplementary Figure S2). These 30-
minute median values were subsequently divided, somewhat arbitrarily, into 5 classes (Table 2.

**Table 2: Classes of soil volumetric water content (VWC) at 2.5 cm depth, where $\theta_{max}$ is the maximum of the 30-minute median values observed during the first hydrological year**

| Class | Category | Criteria |
|---|---|---|
| 1 | Very dry | $\leq 0.1 \times \theta_{max}$ |
| 2 | Dry | $> 0.1 \times \theta_{max}$ & $\leq 0.3 \times \theta_{max}$ |
| 3 | Intermediate | $> 0.3 \times \theta_{max}$ & $\leq 0.7 \times \theta_{max}$ |
| 4 | Wet | $> 0.7 \times \theta_{max}$ & $\leq 0.9 \times \theta_{max}$ |
| 5 | Very wet | $> 0.9 \times \theta_{max}$ |





The temporal pattern of the 5 VWC classes during the first post-fire year is shown in Figure 5, while the corresponding pattern
of the 30-minute median VWC values is given in Supplementary Figure S2. The driest soil conditions (classes 1 and 2)
prevailed during the initial and final periods of this study, from  October to November 2017 and from July to October 2018,
while the wettest conditions (class 5) only occurred occasionally, during March 2018 following intense rainfall (see
Supplementary Figure S3).

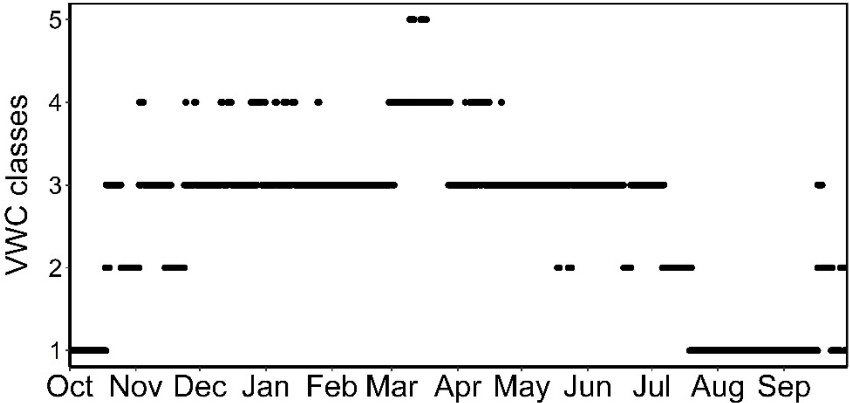


**Figure 5. Volumetric water content (VWC) classes of the topsoil (from dry (1) to wet (5), as defined in Table 2) during the
hydrological year 2017/18. The corresponding 30-minute VWC values are in Supplementary Figure S2.**

### 2.3.6 MAD-Test

In order to eliminate some outliers (spikes) from the data selected for parameterization in the gap-filling procedure, the MAD-
Test (*MAD*: Median Absolute Deviation) was applied. The MAD-Test according to Hoaglin et al. (2000), first applied to $CO_2$
flux data by Papale et al. (2006) and first used for de-spiking raw EC data by Mauder et al. (2013). The MAD-Test identifies
as outlier all values that are outside the following range:

$$median(x) - \frac{q\,MAD}{0.6745} < x_i < median(x) + \frac{q\,MAD}{0.6745} \qquad (5)$$

where the factor of 0.6745 stems from the Gaussian distribution, and $q$ is a threshold value that must be determined depending
on the specific data set.

### 2.3.7 Spike test

The spike test was used in the final stage of data processing (Supplementary Figure S8). While the MAD test is used when the
measured values to be examined scatter only slightly around a mean value, the spike test is used for more scattering data. This
test determines the standard deviation of the entire data set and excludes all values that deviate by a multiple of the standard
deviation. For the spike test, a factor of 3.5 was used as threshold, following Højstrup (1993). The spike test must be carried
out multiple times until the standard deviation hardly changes, which happened after 2–4 times in this study.





## 3 Results

### 3.1 Additional data quality tests

### 3.1.1. Influence of mechanical turbulence

The fact that the bulk of the burnt tree trunks continued upright during the study period raised concerns about their possible impact on turbulence conditions. Therefore, mechanical turbulence was tested according to Foken and Leclerc (2004). The test parameter is the standard deviation of the vertical wind velocity normalized by the friction velocity $\sigma_w/u_*$, and was also used here in the quality flagging of the turbulent data  (Section 2.3.1). The test was carried out with 12,011 30-minute records that were selected for conditions of neutral stratification (-0.2 < $z/L$ < 0.1) and data quality classes 1-8 (i.e. without footprint

selection). The average and standard deviation of $\sigma_w/u_*$ were 1.19 and 0.16, which agreed with available parameterizations (Foken, 2017; Panofsky et al., 1977). The data also confirmed the dependency of the test parameter on stratification (not shown here). The distribution of the test parameter according to wind direction (Figure 6) revealed higher median values for the 30-60° and 60-90° sectors. This could be explained as the typical effect of wind flowing through the tower and coming from the back side of the sonic anemometer, in line with Li et al. (2013). Furthermore, the large patch of eucalypt trees in the 30-60°

sector (Figure 1) could have caused additional turbulence, especially as they re-sprouted vigorously soon after the fire. The method applied for data treatment also flagged data from these sectors. The parameter values for the other wind sectors suggested a tendency for lower median values for the sectors between 210 and 270°, and higher median values for the sectors between 270° and 30°, possibly caused by downhill and uphill flows, respectively (Supplementary Figure S1 for map of slope angles). In overall terms, the values were within the typical range and did not suggest that the standing trunks of either pine or

eucalypt had a relevant impact on data quality.

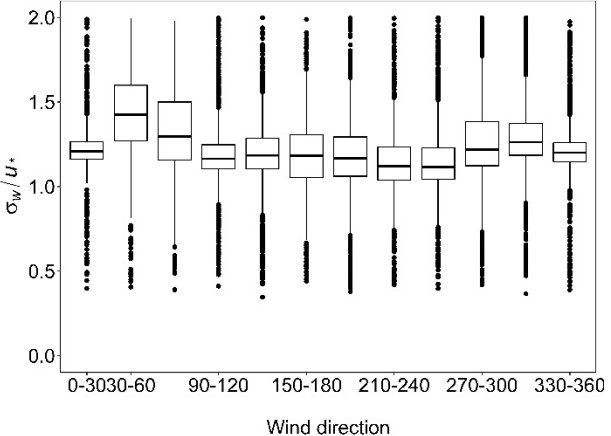

**Figure 6. Box plots of the normalized standard deviation of the vertical wind velocity ($\sigma_w/u_*$) for the individual wind direction sectors.**





### 3.1.2 Energy balance closure


The energy balance, defined as the sum of the turbulent sensible and latent heat fluxes, the net radiation and the ground heat flux, is not fully closed for many turbulent flux sites for multiple reasons that are in most cases not related to measuring errors (Foken, 2008; Mauder et al., 2020). In the case of the present study site, 24 June 2018 (with solar noon 12:38 UTC) was a typical example of a day with mostly clear sky, even if some influence of high clouds was suggested by the comparison of net

radiation and sensible heat flux (Supplementary Figure S4). On average, there was nearly no residual; however, residuals reached values of up to 100 W m$^{-2}$ occurred in the afternoon and even up to about 200 W m$^{-2}$ occurred in the morning  The most likely reason for these discrepancies were errors in the calculation of the ground heat flux (Section 2.3.4). The calculation of ground heat flux from soil temperature and volumetric moisture content may be less appropriate for post-fire condition. More specifically, the present experimental set-up ignored the presence of a – black – wildfire ash layer and may not have fully

captured the soil temperature gradient. This gradient was possibly steep in the first few mm, especially when the soils were dry and still covered by wildfire ash and not yet by vegetation, due to the increased direct insolation combined with a low soil heat capacity and conductance. While the possibility that the net radiation measurements suffered from a slight inclination of the radiometer (in SW direction) cannot be altogether excluded, net radiation did appear to be underestimated. Net radiometers, as the one used in this study (Table 1), that do not measure the four up- and down-welling long- and shortwave radiation

components separately, are well-known to underestimate net radiation (Kohsiek et al., 2007). In the case of the model preceding the one used in this study (i.e., the NR-lite-1), Brotzge and Duchon (2000) reported underestimations of up to 100 W m$^{-2}$ at noon, with a strong sensitivity to the wind speed. The prevalence of negative residual fluxes during the morning was in line with an underestimation of the net radiation due to strong upwelling longwave radiation as a result of high surface temperatures, producing a bias that should be smaller during the afternoon. Because of this possible bias in the closure of the energy balance,

a MAD-Test was applied to the ratio of the turbulent fluxes and the available energy (i.e. net radiation minus ground heat flux), with a factor $q = 0.5$ having been selected as optimal (Section 2.3.6). As shown in Supplementary Figure S5, the gap in energy balance closure amounted to about 10 %, which is within the range of typical values. Therefore, energy balance closure was considered not to pose a significant problem in the calculation of the fluxes.

Turbulent fluxes may also need to be corrected depending on the ratio of sensible and latent heat fluxes, i.e. the Bowen ratio.

In case of a Bowen ratio larger than 1, the sensible heat flux is assumed to be underestimated;  in case of a Bowen ratio below 1, both sensible and latent heat fluxes are assumed to be affected (Charuchittipan et al., 2014; Mauder et al., 2020).  As shown in Supplementary Figure S6, almost all EC measurements under the driest soil conditions (VWC classes 1 and 2)  had a Bowen ratio larger than 1, whereas the same was true for roughly 3 quarters of the measurements under intermediate and wet soil conditions (VWC classes 3 and 4). Therefore, the latent heat flux was not substantially affected and, hence, the $CO_2$ fluxes did

not need further correction.



### 3.2 Gap filling of respiration and assimilation

#### 3.2.1. Respiration

Standard approaches for gap filling were assumed to be less adequate for the present study for two reasons: (i) the marked recovery of the above-ground vegetation in the course of the observation period, in particular from early spring 2018 onwards;
(ii) the important role of soil moisture content in soil respiration fluxes, as is typical for Mediterranean and dry ecosystems (Richardson et al., 2006; Sun et al., 2016). The latter was confirmed by a preliminary analysis of the nighttime NEE fluxes for the different soil VWC classes (not shown but evident in Figure 7), so that the gap filling was done separately for very dry to dry soil conditions (VWC classes 1 and 2), and for intermediate to wet (VWC classes 3 and 4) conditions. Respiration fluxes were determined using only the measurements from 10 pm to 04 am UTC, with more than 80% of the footprint area from the
Maritime Pine stands, and with quality flags 1-6. The selected measurements were subsequently subjected to a MAD-Test, following Papale et al. (2006) and using $q = 0.5$ (Equation 5), The $Q_{10}$ and $E_0$ parameters for the two VWC categories were estimated using the median fluxes of 5 K classes between 10 °C and 30 °C.

In the case of the (very) dry soil moisture conditions, the $Q_{10}$ and $E_0$ parameters were estimated to be 0.154 μmol m$^{-2}$ s$^{-1}$ and 316.6, respectively. The value for $Q_{10}$ was comparatively low (Falge et al., 2001), while $E_0$ was relatively high but still within
the range found in other studies (Reichstein et al., 2005), comparable to that for boreal forests. In the case of the intermediate to wet soil moisture conditions, no realistic median flux values were obtained for the lowest temperature class (10 ± 2.5 °C), even if the flux data were within the detection limit and in spite of the strong data quality tests. Furthermore, the NEE data of the 12.5-17.5 °C and 17.5-22.5 °C temperature classes revealed a suspiciously strong scatter and gave rise to an unrealistically high estimate for E0.Therefore, the abovementioned values of $Q_{10}$ and $E_0$ were used for filling gaps in nighttime NEE fluxes
and for estimating of daytime respiration fluxes, independent of soil moisture conditions. This most likely resulted in an underestimation of the cumulative respiration fluxes presented underneath, as Figure 7 revealed lower nighttime fluxes under very dry to dry soil moisture conditions than under intermediate to wet conditions.

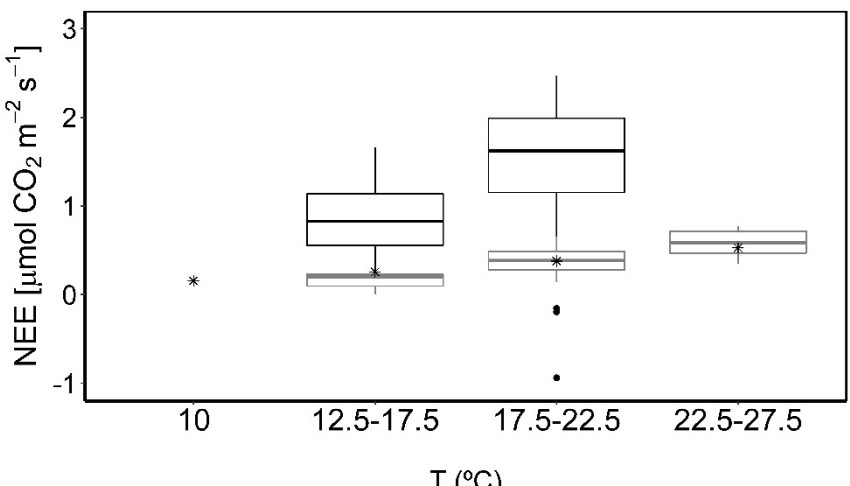

**Figure 7. Box plots of the nighttime NEE fluxes for 5K temperature classes under very dry to dry (VWC classes 1 and 2: grey boxes) and intermediate to wet soil moisture conditions (VWC-classes 3 and 4: black boxes). The asterisks indicate the respiration fluxes calculated following the parameterization of the Lloyd-Taylor function.**

### 3.2.2. Assimilation

Since global radiation was not measured in this study, assimilation was gap filled (Section 2.3.2, Equation 2) using the net radiation corrected with the longwave net radiation for an assumed cloud height of 2-4 km and assuming a low albedo of the surface, following:

$$R_{n-corr} = R_n + \sigma_{SB}[T^4 - (T - 20)^4] \qquad (6)$$

where $\sigma_{SB}$ is the Stefan-Boltzmann constant

According to Falge et al. (2001b) and Hollinger et al. (1994), the factor $a$ in Equation 2 is the linear slope of the assimilation function beginning for a global radiation of 0 W m$^{-2}$ and analogue for this modified net radiation. The slope of the assimilation function, $a$, and the assimilation at radiation saturation, $Q_{c,sat}$, were determined for 5 K binned classes for a data set with footprint > 80% from the pine area, and data quality classes 1-6. The results of this parameterization are summarized in Table 310 3.



**Table 3 Values of the two constants of the Michaelis-Menten assimilation function obtained for the separate 5K classes**

| Temperature | Assimilation | |
|---|---|---|
| | $a$ | $Q_{c\ sat}$ |
| [°C] | [µmol s⁻¹ W⁻¹] | [µmol m⁻² s⁻¹] |
| < 10 | -0.02 | -5.5 |
| 10–15 | -0.02 | -5.0 |
| 15–20 | -0.02 | -5.5 |
| 20–25 | -0.02 | -6.0 |
| 25–30 | -0.02 | -6.0 |
| 30-35 | -0.02 | -5.5 |
| >35 | -0.02 | -5.5 |


### 3.2.3 Generation of the final data set

The flow chart of data processing for the generation of the final data set is shown in Supplementary Figures S7 and S8. Missing data as well as data with quality flag 9, together amounting to 5% of the entire data set, were replaced with estimates computed using the Lloyd-Taylor and Michaelis-Menten functions, following the parameterizations detailed in the two prevision
sections. The same was done for another 5% of the entire data set, comprising the data that did not pass the spike test that was applied to the data with quality flags 1 to 8. The nighttime period - during which assimilation was assumed zero and, hence, just the Lloyd-Taylor function was applied to estimate NEE - was defined as the time between sunset - 15 min and sunrise + 15 minutes because global radiation was not measured. Gap filling of 238 daytime 30-min records was hampered by missing net radiation data, so that they were substituted with interpolated values. The same was done for the 473 estimates from gap
filling that did not pass a second spike test. For periods up to 5h, interpolated values were calculated by linear interpolation between the two values immediately before and immediately after the period; for longer periods, they were computed per time-of-the-day 30-min interval, as the average of the values of the 15 preceding and 15 succeeding days.

### 3.3. Selected cases

Five 3-6 day periods with good footprint conditions were selected to illustrate distinct flux conditions that were identified
during the first post-fire hydrological year (including the first measurement days during September 2017).

### 3.3.1 The role of dew formation

The period from 26-29 September 2017, immediately after the tower became operational, was selected for revealing the role of dew formation on NEE fluxes (Supplementary Figures S9 and S10). By then, no rainfall had occurred after the wildfire (Supplementary Figure S3) and the topsoil was very dry (Figure 5). During this period, the sky was mostly clear, the sensible




heat flux was of the same order as the net radiation, and the Maritime Pine stands generally comprised more than 60 % of the footprint area. The fluxes of both $CO_2$ and NEE (including storage term) were about zero during nighttime and showed an uptake up to -5 μmol m$^{-2}$ s$^{-1}$ during daytime. A substantial emission of $CO_2$ only occurred around sunrise on 28 September 2017, when the relative humidity at the top of the flux tower reached 80-90 % and dew formation took place. The occurrence of dew formation can be inferred from relative humidity in combination with sensible and latent heat fluxes (Foken, 1990).

Dew formation simultaneously produces a positive, upward sensible heat flux due to the heat of condensation, and a negative, downward latent heat flux, while the subsequent evaporation of the dew produces fluxes of the opposite signs. Worth noting, however, is that the observed latent heat fluxes were always below the detection limit of ± 10 W m$^{-2}$ (Mauder et al., 2006) during the early morning hours, reflecting the very dry soil conditions.

The suggestion that the positive NEE flux during the early morning of 28 September 2017 was triggered by dew formation,
was further analyzed by calculating the NEE fluxes with 1-minute time resolution, using the wavelet method (Sect.2.3.3), and comparing them with the relative humidity and the sensible heat fluxes with the same time resolution (Figure 8). The WPL-correction (Webb et al., 1980) was not applied, because it would be very small under the specific conditions and, therefore, would not have noticeably changed the $CO_2$ fluxes.

As shown in Figure 8, relative humidity was about 80% at the top of the flux tower during the early nighttime hours of 28
September 2017, and presumably close to 100% near the ground because of the clear sky and associated temperature gradient. The recorded fluctuations in relative humidity related to fluctuations in sensible heat fluxes and $CO_2$ fluxes. Before 06:00 UTC, however, both fluxes were below their respective detection limits. At around 06:30 UTC, on the other hand, relative humidity increased to 85% and this increase was associated with sensible heat fluxes of up to 20 W m$^{-2}$, clearly in line with the occurrence of dew formation. After 07:00 UTC, relative humidity decreased again to below 80%, creating conditions for
the evaporation of the dew. This dew evaporation was also indicated by negative sensible heat fluxes of up to –30 W m$^{-2}$ between 7.15 and 7.30 UTC, because the evaporation process requires energy. In turn, this peak in negative sensible heat fluxes was accompanied by a peak in upward $CO_2$ fluxes, suggesting that the upward water vapor flux worked as a kind of a pump for $CO_2$ emissions.





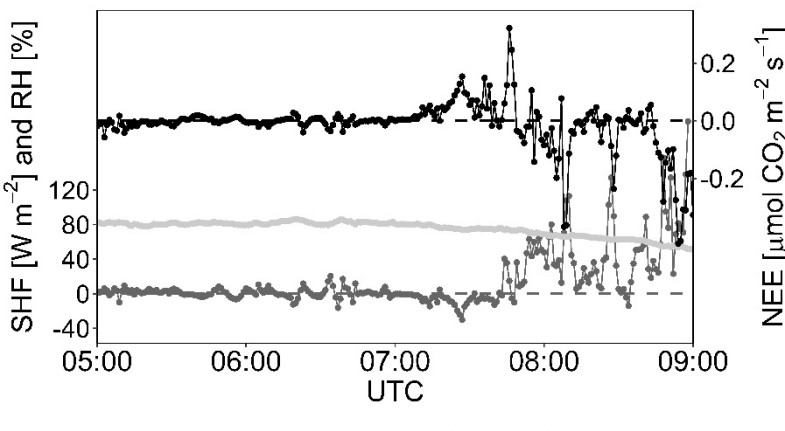

**Figure 8. Relative humidity (RH) and sensible heat (SHF) and NEE fluxes with a 1-min resolution during the morning hours of 28** **September 2017, indicating dew formation followed by evaporation of dew and associated $CO_2$ emission between 06:30 and 07:30** **UTC.**

### 3.3.2 The role of the first rainfall events after the wildfire

The first post-fire rainfall events occurred more than two months after the wildfire, between 17 to 22 October 2017 and significantly increased soil VWC (Supplementary Figures S2 and S3). The bulk of this rainfall occurred during the night from 17 to 18 October 2017 (8.4 mm) and around noon on 20 October 2017 (3.2 mm). During this 6-day period, the footprint area generally consisted for more than 80% of the Maritime Pine stands, cloudy conditions prevailed (in spite of sunny periods on 17, 18, and 22 October 2017), and the latent heat flux contributed markedly to the energy exchange (Bowen ratio of about 1; Supplementary Figure S11) because of the high relative humidity (exceeding 90% during rainfall). With the onset of the autumn rainfall, the ecosystem started to be a source of $CO_2$ but the fluxes decreased again on 22 October 2017. Worth noting was that the second, smaller rainfall event of 20 October 2017 seemed to have a greater impact on $CO_2$ emissions than the first event of 17-18 October 2017. The large scatter in NEE fluxes observed during some periods could be explained by conditions of low turbulence and the generally low fluxes.

The role of rainfall periods in NEE fluxes during the initial post-fire window-of-disturbance was also evidenced by the cumulative NEE values from 1 October to 31 December 2017 (Figure 9). After an initial period of net assimilation, three marked peaks in net $CO_2$ emissions occurred that were associated with periods of intense rainfall during mid-October, early and late November 2017. By contrast, intense rainfall periods during early and especially also late December 2017 only had minor impacts on net $CO_2$ emissions. This was probably due to the lower temperatures, ranging from 5 to 15°C during daytime.





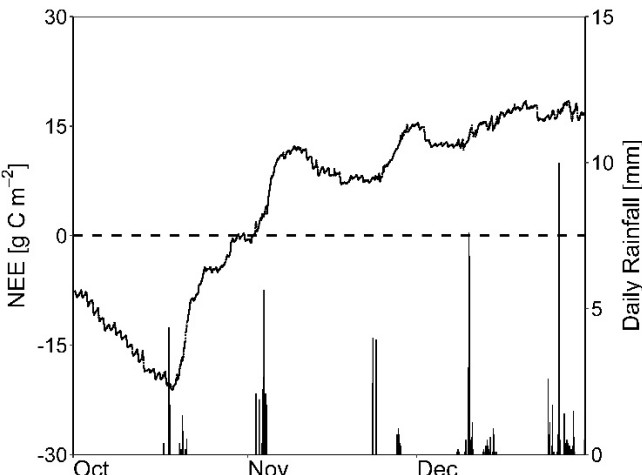


**Figure 9. Cumulative NEE fluxes and 30-min rainfall during the initial window-of-disturbance, from 01 October to 31 December 2017.**

### 3.3.3 The role of woodland type and (antecedent) rainfall during summer conditions

Energy and NEE fluxes from the Maritime Pine stands under dry conditions during the first post-fire summer are illustrated in Figure 10. During the selected, rainless 6-day period from 4 to 9 September 2018, the footprint area generally consisted for more than 80% of the pine stands, while topsoil VWC was consistently very dry (class 1; Figure 5), reflecting the less than 1 mm of antecedent rainfall over the preceding 4-week period (Supplementary Figure S3). The diurnal pattern in NEE fluxes was similar to the average monthly trends reported by Serrano-Ortiz *et al.* (2011) for a pine stand that had burnt 4 years earlier

and had not been intervened afterwards, and in particular, to that of June because of the greater contrast between daytime and night-time fluxes.

   A second rainless summer-2018 period was selected to analyse energy and NEE fluxes from the eucalypt patches located to the east of the tower. Even though this 4-day period was about one month earlier, from 6 to 9 August 2018, topsoil VWC was equally very dry and antecedent rainfall over the preceding 4-week period was equally less than 1 mm. NEE fluxes under

summer-2018 conditions did not differ conspicuously between the eucalypt (Supplementary Figure S12) and Maritime Pine stands (Figure10), neither in terms of diurnal patterns nor in terms of measured values. As to be expected, sensible heat fluxes did differ markedly, being clearly higher during early August than early September. The same was true for the Bowen ratio, attaining an average value as high as 5.4 over the 6-9 August 2018 period, as opposed to 2.7 over the 4-9 September 2018 period.

A 3-day period during early July 2018 was selected to examine how summer-2018 NEE fluxes from the Maritime Pine stands (comprising > 80% of the footprint area) were affected by (antecedent) rainfall (Supplementary Figure S13). Two minor rainfall events (defined here as periods preceded and succeeded by at least 3h without rainfall) occurred on 1 July 2018. The



first one started at 21.00 UTC on 30 June and ended at 02.30 UTC on 1 July and amounted to 2.0 mm, and the second lasted from 14.00 UTC to 15.30 UTC on 1 July and amounted to 0.4 mm (Supplementary Figure S3). These rainfall events lead to a

minor increase in topsoil VWC (Supplementary Figure S2). Arguably, the main contrast with the early September period was the antecedent rainfall, amounting to 40,3 mm as opposed to 0.1 mm over the preceding 14 days. This contrast was also reflected in topsoil moisture conditions, which were moderate (VWC class) during early-July as opposed to very dry (VWC class 1) during early-September. The NEE fluxes during the early July-period, however, did not differ markedly from those of the early-September period. Apparently, neither assimilation nor respiration processes suffered from serious moisture

limitations by early September, in spite of the very dry conditions of the topsoil.

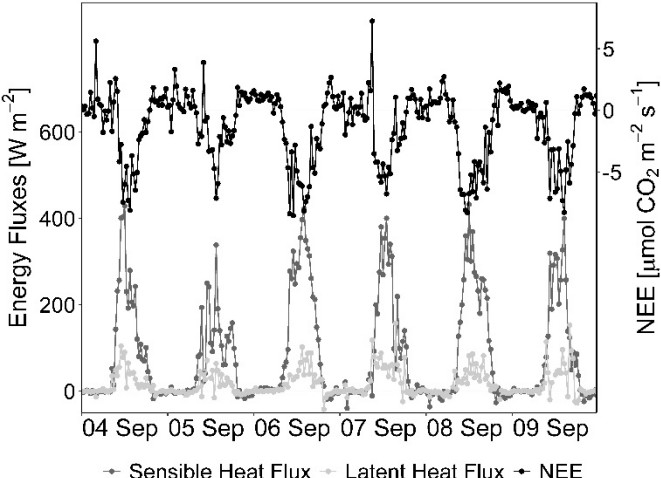

**Figure 10. Energy and NEE fluxes from the Maritime Pine stands during a rainless period towards the end of the first post-fire hydrological year, from 04 to 09 September 2018.**

**3.4 Cumulative Carbon dioxide fluxes**

The cumulative $CO_2$ fluxes over the first hydrological year following the wildfire (01 October 2017 – 30 September 2018) are shown in Figure 11. No distinction was made between the fluxes from the Maritime Pine stands and those from the eucalypt stands for two reasons. First, because the pine-stand fluxes were predominant, given the prevailing WNW to N wind directions; second, because the fluxes from the two forest types appeared to be similar during summer 2018 (Section 3.3.3). In terms of NEE patterns, five different periods could be distinguished during this first post-fire year. During the immediate post-fire

period, which ended with the first rainfall event on 17 October 2017, the burnt area acted as a carbon sink, even if just a small one. During the ensuing period, which ended in mid-December 2017, the burnt area functioned as a carbon source, especially following periods of intense rainfall. During the coldest period from mid-December 2017 until the end of January 2018, NEE fluxes were close to zero. With the onset of warmer temperatures during early February 2018, followed by a (practically) rainless February month, the area started to become a small carbon sink. This period continued during the next two, rainiest



months, during which short intervals occurred when respiration was the dominant process. Finally, from May 2018 onwards, the area was a marked carbon sink, with assimilation clearly prevailing over respiration.

The cumulative assimilation over the first post-fire hydrological year was roughly twice the cumulative respiration. This was remarkable since forest carbon flux studies have generally found both to be of the same order of magnitude (e.g. Luyssaert et al., 2010). This discrepancy between assimilation and respiration resulted to a large extent from the last of the five above-

mentioned periods, starting in May 2018. A possible reason for the discrepancy was the impossibility to parameterize respiration under intermediate and wet soil conditions and, hence, that respiration was possibly underestimated. However, the number of gap-filled data was very low.

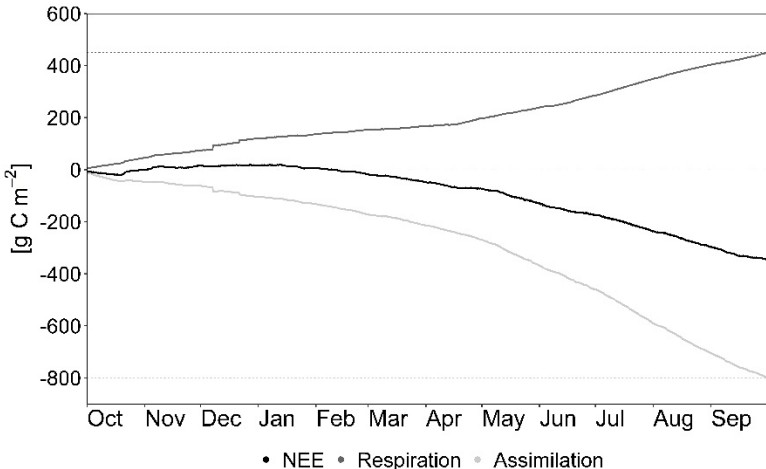

**Figure 11. Cumulative NEE, assimilation and respiration fluxes during the first hydrological year after wildfire, from 01 October**
**2017 to 30 September 2018.**

## 4. Discussion

The following discussion focuses on the three main novelties of this study, also because of the lack of comparable EC studies of ecosystem recovery during the initial stages following wildfire.

### 4.1. Data analysis

The present data set was less suited for the standard procedure of data quality assessment using a threshold of friction velocity (Goulden et al., 1996) and, hence, standardized data analysis routines as used in networks such as ICOS and NEON (Metzger et al., 2019; Rebmann et al., 2018). NEE fluxes tended to be very low during the first year after the wildfire and wind speeds were generally low, not exceeding 3 m s$^{-1}$. To address these particular conditions, a specific procedure was developed in the present study, based on data quality flagging (Ruppert et al., 2006). It allowed limiting the need for gap filling to about 5 % of

the data, while gap filling of up to 30 % of the data is common for the fraction velocity-based procedures. The present procedure, however, required thorough data analysis, involving not only repeated MAD and spike tests and modelling of the



footprint area but also assessing the possible influence of the standing burnt tree trunks on mechanical turbulence. Furthermore, an exploratory analysis of the closure of the energy balance was carried out, including because ground heat fluxes were not measured directly in this study. The energy balance closure proved acceptable, not raising major concerns about the correctness

of the measured carbon fluxes. In addition, the Bowen ratio was typically in the range of 1 to 5, thereby guaranteeing that the gaps in  energy balance closure did not markedly influence the carbon fluxes and that these fluxes did not require correction for such gaps (Charuchittipan et al., 2014). Gap filling itself was based on careful parameterization of the Lloyd-Taylor function, in particular by taking into account soil moisture as a key factor in the respiration of Mediterranean and dry ecosystems (Richardson et al., 2006; Sun et al., 2016). The nighttime NEE data also revealed this importance of soil moisture,

but only allowed a reliable parameterization of the Lloyd-Taylor function for very dry to dry soil moisture conditions and not for intermediate to wet conditions. The latter could be due to the fact that these intermediate and wet conditions included three of the five periods that were distinguished in terms of NEE fluxes (Section 3.4), with fluxes ranging from practically zero during early winter (mid-December 2017 to end of January 2018) to their highest values during late spring (May to June 2018). Finally, the potential of the wavelet method as a complementary tool to analyse specific short-term events with an elevated

temporal resolution was demonstrated, as it provided crucial insights into $CO_2$ fluxes during and following dew formation.

### 4.2. Respiration fluxes upon wetting by dew formation

Dew formation has been reported for many climate types, affecting, amongst others, microbial activity during rainless periods (Agam and Berliner, 2006; Gliksman et al., 2018; Verhoef et al., 2006). To the best knowledge of the authors, dew formation

had not yet been observed in burnt areas. Its impact on ecosystem respiration differed fundamentally from the Birch effect that Sánchez-García et al. (2020) observed in the same burnt area as studied here and equally before the occurrence of post-fire rainfall (i.e. on 17 october 2017 in case of the site with wildfire ash). Sánchez-García et al. (2020) reported the highest soil effluxes immediately after stopping the simulated rainfall (after 10 min), but inferred, based on wetting experiments with the same soils under laboratory conditions, that peak values had occurred even earlier. The short duration of the peak was argued

to suggest that the Birch effect resulted from the displacement of $CO_2$-rich air in soil and especially ash pores by infiltrating water (degassing), including because of the likely suppression of microbial activity due to the still recent sterilization by the fire. For the same reasons, the dew-induced $CO_2$ efflux observed in this study was probably due to a physical process rather than to microbial activity. This process, however, differed from the displacement of ash-soil air by infiltrating water in the sense that the respiration flux only started some half an hour after the dew formation, with the onset of the evaporation of the

dew. The observed water vapor flow from the soil surface was large enough to generate a pumping effect with vertical wind velocities at the surface in the order of $10^{-4}$ m s$^{-1}$ (Webb et al., 1980), on the one hand, and, on the other, the amount of dew was sufficient to explain the $CO_2$ efflux between 07:01 and 07.41 UTC. This $CO_2$ efflux amounted to 4.94 mg m$^{-2}$ or 2.50 cm$^3$ m$^{-2}$. The sensible heat flux during this 40-min period was 17.3.10$^3$ J m$^{-2}$, i.e., involved enough energy to evaporate the 7.03 g m$^{-2}$ or 7.02 cm$^3$ m$^{-2}$ of dew, which, in turn, is equivalent to 0.41 m$^3$ m$^{-2}$ of $CO_2$ gas (see Foken et al. (2020) for the temperature-



dependent physical parameters). The occurrence of this pumping effect rather than the degassing effect observed by Sánchez-García et al. (2020) was probably due to the comparatively small amount of dew water (0.007 vs. 25 mm of simulated rain), combined with the presence of a considerable wildfire ash layer. The inter-patch ash load determined at the 5 transect points on 7 September 2017 averaged 2.21 g m$^{-2}$, with a minimum of 762 g m$^{-2}$. This ash layer will have easily absorbed the small amount of dew water, as wildfire ash has an elevated water storage capacity (Balfour and Woods, 2013; Leighton-Boyce et

al., 2007). Probably, the wetting of the ash layer was limited to its immediate surface, not causing significant degassing of ash pores underneath. This wetting might have created a kind of a seal, even if perhaps a spatially heterogeneous one as Sánchez-García et al. (2020) reported that almost 50 % of the pine ash was severely to extremely water repellent. The subsequent evaporation would then have broken this seal and/or simply pumped out part of the $CO_2$ stored in the underlying ash pores. Further research is need to clarify to which extent the emitted $CO_2$ originated from a rapid restoration of microbial respiration

caused by microbial biomass growth and the activation of extracellular enzymes, as has been observed after the first post-fire rainfall events (Fraser et al., 2016; Waring and Powers, 2016).

### 4.3. Cumulative NEE fluxes

The discussion of the cumulative NEE fluxes of this study is seriously hampered by the limited number of post-fire EC studies and, in particular, by the existence of just one prior EC study that monitored a large part of the first post-fire year

(Supplementary Table S1). This latter study, of Sun et al. (2016), found that a Eucalypt spp. woodland in southern Australia was a net carbon source for a considerably longer post-fire period than the present site, i.e. until the 15$^{th}$ instead of the 5$^{th}$ month after fire. This delay could be due to the much drier, semi-arid climate conditions together with low soil nutrient availability, resulting in a reduced pre-fire NEP (< 100 g C m$^{-2}$ y$^{-1}$) of the patchy, low-stature vegetation. The net carbon emissions during the first three monitoring months of Sun et al. (2016), however, did not differ widely from the cumulative

NEE fluxes observed in this study over the 2 months following the first post-fire rainfall events (mid-October to mid-December 2017). The former ranged from 11 to 19 g C m$^{-2}$ month$^{-1}$ for post-fire months 4 to 6, whereas the latter averaged about 20 g C m$^{-2}$ month$^{-1}$. The other post-fire EC studies suggested that re-establishment of carbon sink function after fire took at least 1 to 9 years (Amiro et al. (2006): >1 year; Dadi et al. (2015): >2 years; Serrano-Ortiz et al. (2011): <4 years; Mkhabela et al. (2009): >6 years; Dore et al. (2008): >9 years).

Comparison of the cumulative annual NEE flux of this study with those of prior EC studies in burnt woodlands and/or unburnt pine areas (summarized in Supplementary Tables S1 and S2) showed that the present cumulative NEE of -290 g C m$^{-2}$ y$^{-1}$ over the first-post-fire year differed least from that reported by Moreaux et al. (2011) for their 4-year old Maritime Pine plot (-243 g C m$^{-2}$ y$^{-1}$). The annual NEE of a second, intervened plot studied by Moreaux et al. (2011), however, was much lower (-65 g C m$^{-2}$ y$^{-1}$). The authors attributed this to the rapid growth of shrubs and herbaceous species following the weeding and thinning,

possibly even compensating a decrease in GPP by the pines due to the thinning. This intervened plot had also been studied earlier by Kowalski et al. (2003), likewise showing that the undergrowth species started fixating carbon just a few months after the clear cutting of the original 50-year old Maritime Pine stand. Shrub and herbaceous species should also explain the bulk



of the GPP at the present site, as they summed an average cover of 97% by mid-September 2018 as opposed to an average tree cover of 13% between Maritime Pine and Eucalypt (Supplementary Table S3).

Even more unexpected than the rapid recovery of the carbon sink function at the present site was the net carbon assimilation observed during the immediate post-fire period, until the first post-fire rainfall events of mid-October 2017. The net assimilation was about 1.0 g C m$^{-2}$ d$^{-1}$, i.e. between the rates during the other two periods with net assimilation (February-April 2018: 0.6 g C m$^{-2}$ d-1; May-October 2018: 1.8 g C m$^{-2}$ d$^{-1}$). Unlike the two assimilation periods from early 2018 onwards, this 2017 assimilation period was difficult to link to the recovery of the understory vegetation for two main reasons: (i) the

understory vegetation was fully consumed by the fire; (ii) the recovery of the understory vegetation was still very reduced by early January, as illustrated for three key resprouter shrub species in Supplementary Figure S14. Possibly, this immediate post-fire photosynthetic activity originated from various patches of pines with scorched crowns to the northeast and south of the EC tower and/or from re-sprouting eucalypts, in particular the 4 individual trees near the tower and/or the patch to the east of it.

**5. Conclusions**

The main conclusion of this first study into CO$_2$ fluxes following wildfire over the first post-fire hydrological year were:

(i) a specific data analysis procedure including data quality flagging, MAD and spike testing, footprint analysis, soil-moisture-dependent gap filling and assessment of mechanical turbulence had to be developed because of the very low fluxes and

prevailing wind speeds below 3 m s$^{-1}$, but allowed to reduce the need for gap filling to just about 5 % of the data;

(ii) the use of the wavelet method for the determination of turbulent fluxes with a 1-minute time resolution proved to be extremely helpful for a detailed analysis of the role of dew formation on soil respiration;

(iii) the cumulative NEE fluxes during the first hydrological year after a wildfire that occurred in August 2017 revealed an intricate temporal pattern that could be divided into five phases. The first phase (first half of October 2017) and the last two

phases (from early February 2018 onwards) were (mainly) governed by assimilation, the second (mid- October to mid-December 2017) was dominated by soil respiration that was closely linked to the first post-fire rainfall events, and the third phase (mid-December 2017 to early February 2018) had negligible fluxes;

(iv) the carbon sink function of this Maritime Pine-dominated area was re-established within less than half a year after the wildfire, mainly due to the recovery of the understory vegetation of both resprouter and seeder species;

(v) dew formation during the rainless, immediate post-fire period produced a noticeable soil carbon efflux that was linked to dew evaporation and not to instantaneous degassing due to wetting.

**The Supplement related to this article is available online at doi: …………-supplement**



*Code and data availability.* The program for the calculation of the EC data is available (Mauder and Foken, 2015). The NEE, Assimilation and Respiration data after gap-filling are available on Oliveira et al. (2020). Other data can be requested by email to bruna.oliveira@ua.pt.

*Authors contribution.* B.R.F. Oliveira was responsible for setting-up and operating the flux tower, carried out the analysis of
the EC data, prepared the tables and figures and drafted most sections; C. Schaller carried out the wavelet analysis, analysed its results and drafted the respective section; J.J. Keizer wrote the grant proposal, coordinated the project work, created the vegetation map, analysed the soil moisture data, and drafted the respective sections; T. Foken was scientific adviser of the project, selected instrumentation of the flux tower, defined site selection criteria, outlined and supervised data analysis, conceptualized the structure of the paper and directed its write-up. All authors actively contributed and agreed with the final
version of the paper.

*Competing interests.* The authors declare that they have no conflict of interest.

*Acknowledgements.* This work was supported by the project FIRE-C-BUDs (PTDC/AGR-FOR/4143/2014 - POCI-01-0145-
FEDER-016780), funded by the national Foundation for Science and Technology of Portugal (FCT/MEC) with co-funding by the FEDER, within the PT2020 Partnership Agreement and Compete 2020 and with additional support by CESAM, through the strategic project UID/AMB/50017, funded by the FCT/MEC with co-funding by the FEDER, within the PT2020 Partnership Agreement and Compete 2020. We would like to acknowledge the help of Penelope Serrano-Ortiz from the University of Granada and the colleagues from the FIRE-C-BUDs project Isabel Campos, João Pedro Carreira (uav
photography for Figure 1, and DSM and slope angle maps of Supplementary Figure S1), Mário Cerqueira, Oscar González-Pelayo, Cláudia Jesus, Paula Maia (vegetation relevees for Supplementary Table S3), Martinho Martins, Luísa Pereira, Glória Pinto, Casimiro Pio, and Alda Vieira. Furthermore, we would like to thank Nuno Costa, António Martins, José Pedro Rodrigues and Sr. Guilherme for their help in preparing and mounting the flux tower. We also thank Renato Santos, D. Benvinda and Sr. José for allowing the installation of the tower on their land.
This publication was funded by the German Research Foundation (DFG) and the University of Bayreuth within the funding program Open Access Publishing.

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
