# Peer review of "Estimating immediate post-fire carbon fluxes using the eddy-covariance technique"

_Biogeosciences, 2020_

## Referee Comment (RC1) · Tarek El-Madany (Referee) · 15 Sep 2020

The manuscript entitled "Estimating immediate post-fire carbon fluxes using the eddy-covariance technique" presents a unique data set of carbon-, water-, and energy fluxes measurements 43 days after a wildfire in Portugal. The ecosystem is/was a Maritime Pine with some Eucalyptus stands inside which were mainly burned and only the trunks of the trees remained. The authors explain broadly their data quality control scheme how data were filtered, selected and gap filled. Based on the data they represent cumulative fluxes of NEE GPP and Reco. Additionally, they focus based on one event on the interactions of dew and ashes with respect to carbon dioxide fluxes. Overall the manuscript is well written, but it could need some polishing on the figures as well on the text where sometimes method parts are in the results and discussion

parts are in the results. I think this should be cleaned up. Overall, the manuscript is worth to publish especially because we don't have many studies presenting data from ecosystems shortly after a fire disturbance and its recovery. My main concern with the manuscript is that the authors interpret a lot into the eddy covariance data without having the right measurements to back it up. See my comments for details. Further, I think the u*-threshold estimation and removal of data with low u* values is nothing to debate about as the eddy covariance technique is not working under those conditions and this must be accounted for in the data processing. For further details see the attached pdf.

Please also note the supplement to this comment:
https://bg.copernicus.org/preprints/bg-2020-312/bg-2020-312-RC1-supplement.pdf

---

## Short Comment (SC1) · 29 Sep 2020

The authors would like to thank Tarek El-Madany for his helpful comments, which helped us to resolve some ambiguities in the manuscript. Please see the detailed discussion of each comment in the attached file.

Please also note the supplement to this comment:
https://bg.copernicus.org/preprints/bg-2020-312/bg-2020-312-SC1-supplement.pdf

---

## Referee Comment (RC2) · Anonymous Referee #2 · 9 Oct 2020

The paper by Oliveira et al., provides exceptional and very valuable information about the CO2 flux behavior immediately after a wildfire, and therefore I encourage its publication. However, the paper, in its present form is difficult to follow and should be re-structured before its publication. Results section 3.1 and 3.2 should be moved to methodology. And the section 3.3 about results should not have references. References (and its arguments) should be moved to the discussion section. See my specific comments below:

The objective mentioned at the end of the introduction section is not a real objective. The "in-depth analysis of the obtained EC data" is the way (the method) to analyze the behavior of CO2 and water vapor fluxes and its dominant factors immediately after a wildfire (your objective). Another objective of your paper could be the optimization of

the quality tests for EC data to correctly interpret the obtained fluxes immediately after a wildfire.

Table 1. I think there is a mistake about the frequency of sample for net radiation. It should be 0.02Hz. This info is correctly written in the above paragraph.

Ln 139-140 "For the cumulative fluxes over the first post-fire year, all EC data with quality classes 1-8 were combined with gap-filled data" What about the footprint area? Did you also selected footprint areas that consisted for more than 80 % of the Maritime Pine stands?

Section 2.3.1. To include the % of missing half-hourly flux data due to measurement failures or rejection after the data quality check could be a very interesting information. This info can be divided into daytime and nighttime data.

Ln 109, A parenthesis after (table 2 is missing. I would located sections 2.3.6 and 2.3.7 before section 2.3.4 because they are also related to EC measurements.

Ln 223 "The test was carried out with 12,011 30-minute records that" There is something wrong in the numbers.

Section 3.1 "Additional data quality test" should be section 2.4.

Section 3.1.2: Since the objective of this study is not to investigate the closure of the energy balance, I would recommend to remove this subsection and to include a sentence in section 2.3.1 with the % of the gap in the energy balance closure (that is in the range reported by most EC sites). If the authors consider that part of this subsection must appear in the manuscript, just move it into discussion section (4.1 data analisys)

The first paragraph for section 3.2.1 is "methodology" not results. Please, move this paragraph to section 2.3.2. What is more, the second paragraph is mostly discussion. Figure 7 is a result, but should be better explained in the text in order to show its relevance.

Section 3.2.2 is again "methodology" not results. Please, move this paragraph to section 2.3.2. Again, despite table 3 is a result, should be better explained in the text in order to show its relevance.

Section 3.2.3 should be also moved to "methodology" section. Ln 336 Please include information in the methodology section about the storage term.

I would recommend to move the figure S10 into section 3.3.1. The inclusion of Figure S10 (maybe it is not necessary to include the four days) would help to the lector to better understand the Figure 8. I would also improve the Figure S10 (and next) including the 0 Y line for NEE and the time in the X axis. The period showed in Figure 8 can be shadow in Figure S10.

The measured $CO_2$ uptake in September and October 2017 should be due to the presence of plant cover in the studied area. Do you have some pictures to test it? Otherwise, you should provide another explanation, for the $CO_2$ uptake in September and October 2017 (Eucalipts?).

Section 4.1. Just curiosity..., did you try to compare the cumulative NEE using your procedure for rejecting data and filling gaps and with the "standard procedure" available in https://www.bgc-jena.mpg.de/bgi/index.php/Services/REddyProcWeb?

---

## Author Comment (AC1) · 13 Oct 2020

Authors' discussion of the comments **bg_2020-312-RC2**

The authors would like to thank Reviewer #2 for his helpful comments. By taking them into consideration, the authors think that they will also clarify some of the misunderstandings of Reviewer #1.

To address the comments in the document *bg_2020-312-RC2*, the authors first copy the exact comment by the reviewer, add numbers to order each comment and make cross-referencing easier, and format in grey and Italic. The answers given by the authors are in black after each comment. In the end of the document, a list of References that support the answers was added.

**General comments**

*The paper by Oliveira et al., provides exceptional and very valuable information about the $CO_2$ flux behavior immediately after a wildfire, and therefore I encourage its publication. However, the paper, in its present form is difficult to follow and should be re-structured before its publication. Results section 3.1 and 3.2 should be moved to methodology. And the section 3.3 about results should not have references. References (and its arguments) should be moved to the discussion section. See my specific comments below.*

In fact, it is not easy to decide whether sections 3.1 and 3.2 should be assigned to the "methodology" section or to the "results" section, since some steps in the data processing required special analyses beyond what is standard practice. In the end, however, the authors agree to follow the arguments of Reviewer #2 to give the manuscript a clearer structure. Consequently, some results in Section 3.3 related to other sources will be included in the discussion.

*The objective mentioned at the end of the introduction section is not a real objective. The "in-depth analysis of the obtained EC data" is the way (the method) to analyze the behavior of $CO_2$ and water vapor fluxes and its dominant factors immediately after a wildfire (your objective). Another objective of your paper could be the optimization of the quality tests for EC data to correctly interpret the obtained fluxes immediately after a wildfire.*

The authors agree with revising the formulation of the objectives of the manuscript, also with regard to the notes of Reviewer #1, to emphasize the efforts for an adequate data analysis under the conditions immediately after a wildfire. The authors will take these comments of the reviewers into account in the revised version of the manuscript.

**Specific comments**

1. *Table 1. I think there is a mistake about the frequency of sample for net radiation. It should be 0.02Hz. This info is correctly written in the above paragraph.*

The net radiation was sampled at 20 Hz. Due to limitations in the data loggers' channels, the net radiometer had to be connected to a "fast" channel so that averaging was carried out in the same way as for "slow" data. The authors will clarify this in the final version of the manuscript.

2. *Ln 139-140 "For the cumulative fluxes over the first post-fire year, all EC data with quality classes 1-8 were combined with gap-filled data"*
   *What about the footprint area? Did you also selected footprint areas that consisted for more than 80 % of the Maritime Pine stands?*

The authors will clarify the text with a short remark at this point. The complete explanation is given in Section 3.3.3.

> 3. Section 2.3.1. To include the % of missing half-hourly flux data due to measurement failures or rejection after the data quality check could be a very interesting information. This info can be divided into daytime and nighttime data.

All relevant information is contained in Supplementary material Figure S8 but the authors will insert a percentage value in the manuscript.

> 4. Ln 109, A parenthesis after (table 2 is missing. I would located sections 2.3.6 and 2.3.7 before section 2.3.4 because they are also related to EC measurements.

The authors thank the reviewer for the correction regarding the parenthesis. Following the re-structuring proposed and acknowledged in the general comments, Section 2 of the final manuscript will be reorganized.

> 5. Ln 223 "The test was carried out with 12,011 30-minute records that" There is something wrong in the numbers.

The authors do not perceive what could be wrong here. A total of 17 760 records were available. For the mechanical turbulence test, 12 011 records with nearly neutral stratification were used, as specified in the sentence:

L223-L225: The test was carried out with 12,011 30-minute records that were selected for conditions of neutral stratification (-0.2 < z/L < 0.1) and data quality classes 1-8 (i.e. without footprint 225 selection).

This data selection is also shown in Figure 2 in the authors' answer to Reviewer # 1.

> 6. Section 3.1 "Additional data quality test" should be section 2.4.
> 7. Section 3.1.2: Since the objective of this study is not to investigate the closure of the energy balance, I would recommend to remove this subsection and to include a sentence in section 2.3.1 with the % of the gap in the energy balance closure (that is in the range reported by most EC sites). If the authors consider that part of this subsection must appear in the manuscript, just move it into discussion section (4.1 data analisys).
> 8. The first paragraph for section 3.2.1 is "methodology" not results. Please, move this paragraph to section 2.3.2. What is more, the second paragraph is mostly discussion. Figure 7 is a result, but should be better explained in the text in order to show its relevance.
> 9. Section 3.2.2 is again "methodology" not results. Please, move this paragraph to section 2.3.2. Again, despite table 3 is a result, should be better explained in the text in order to show its relevance.
> 10. Section 3.2.3 should be also moved to "methodology" section.

Please see answer to general comments. The authors appreciate the specific comments 6-10 and will take these in due consideration when re-structuring the manuscript.

> 11. Ln 336 Please include information in the methodology section about the storage term.

The authors will include the information about the storage term in Section 2.

> 12. I would recommend to move the figure S10 into section 3.3.1. The inclusion of Figure S10 (maybe it is not necessary to include the four days) would help to the lector to better understand the Figure 8. I would

*also improve the Figure S10 (and next) including the 0 Y line for NEE and the time in the X axis. The period showed in Figure 8 can be shadow in Figure S10.*

The authors agree with inserting in Figures S9 and S10 a shadow band for the 4 hours shown in Figure 8. The width is then about 6 mm. Due to the time resolution of Figures S9 and S10, very little will be visible, except for the daily cycle. The authors will show the Figures S9 and S10 in the supplement. Also, the zero line will be added

13. *The measured CO2 uptake in September and October 2017 should be due to the presence of plant cover in the studied area. Do you have some pictures to test it? Otherwise, you should provide another explanation, for the CO2 uptake in September and October 2017 (Eucalipts?).*

The authors discussed this question in detail in the answer to Reviewer #1, and also provided a orto-photomap showing the pine stands with scorched crowns and the 4 individual eucalypts. The authors will convert this discussion into additional text for the manuscript, especially elaborating on which of the possible explanations that were originally postulated (uptake by dying pine crowns or by resprouting plants, especially eucalypt) is most likely.

14. *Section 4.1. Just curiosity..., did you try to compare the cumulative NEE using your procedure for rejecting data and filling gaps and with the "standard procedure" available in https://www.bgc-jena.mpg.de/bgi/index.php/Services/REddyProcWeb?*

The processing of the measured data was carried out according to international standards (Aubinet et al. 2012) with a software package that has been compared internationally several times (Fratini and Mauder 2014). The differences in the definition of the criteria for gap-filling (data quality or u* criterion) were shown in the literature (Ruppert et al. 2006). The application of the u* criterion would result in almost 50 % of the data (instead of 10 %) having to be replaced by modelling. Therefore, there is no need to compare our results with the software package that is "standard" at the Max Planck Institute Jena (Wutzler et al. 2018), as suggested by Reviewer #1. What is certainly recommended is to compare with the software routines used in different international programs, which are standard there, just like the author (TF) did 15 years ago (Mauder et al. 2008). For such a comparison, however, the existing data set is not very suitable, since in addition to biogenic processes, chemical processes linked to the presence of wildfire ashes must also be taken into account while the observed flows were generally very small and near the detection limit.

**References:**

Aubinet M, Vesala T and Papale D (eds) (2012) Eddy Covariance: A Practical Guide to Measurement and Data Analysis. Springer, Dordrecht, Heidelberg, London, New York, 438 pp.

Fratini G and Mauder M  (2014) Towards a consistent eddy-covariance processing: an intercomparison of EddyPro and TK3. Atmos Meas Techn. 7:2273-2281. https://doi.org/10.5194/amt-7-2273-2014

Mauder M, Foken T, Clement R, Elbers J, Eugster W, Grünwald T, Heusinkveld B and Kolle O  (2008) Quality control of CarboEurope flux data - Part 2: Inter-comparison of eddy-covariance software. Biogeoscience. 5:451-462.  https://doi.org/10.5194/bg-5-451-2008

Ruppert J, Mauder M, Thomas C and Lüers J  (2006) Innovative gap-filling strategy for annual sums of $CO_2$ net ecosystem exchange. Agric For Meteorol. 138:5-18. https://doi.org/10.1016/j.agrformet.2006.03.003

Wutzler T, Lucas-Moffat A, Migliavacca M, Knauer J, Sickel K, Šigut L, Menzer O and Reichstein M (2018) Basic and extensible post-processing of eddy covariance flux data with REddyProc. Biogeosci. 15:5015-5030.  https://doi.org/10.5194/bg-15-5015-2018

---

## Author Comment (AC2) · 18 Oct 2020

Authors' discussion of the comments **bg_2020-312-RC1**

The authors would like to thank Tarek EI-Madany for his helpful comments, which helped us to resolve some ambiguities in the manuscript.

To address the comments in the document *bg_2020-312-RC1*, the authors first copy the exact comment by the reviewer, add numbers to order each comment and make cross-referencing easier, and format in grey and Italic. The answers given by the authors are in black after each comment. In the end of the document, a list of References that support the answers was added.

**Major comments**

1. *The manuscript entitled "Estimating immediate post-fire carbon fluxes using the eddy-covariance technique" presents a unique data set of carbon-, water-, and energy fluxes measurements 43 days after a wildfire in Portugal. The ecosystem is/was a Maritime Pine with some Eucalyptus stands inside which were mainly burned and only the trunks of the trees remained. The authors explain broadly their data quality control scheme how data were filtered, selected and gap filled. Based on the data they represent cumulative fluxes of NEE GPP and Reco. Additionally, they focus based on one event on the interactions of dew and ashes with respect to carbon dioxide fluxes.Overall the manuscript is well written, but it could need some polishing on the figures as well on the text where sometimes method parts are in the results and discussion parts are in the results. I think this should be cleaned up. Overall, the manuscript is worth to publish especially because we don't have many studies presenting data from ecosystems shortly after a fire disturbance and its recovery. My main concern with the manuscript is that the authors interpret a lot into the eddy covariance data without thaving the right measurements to back it up. See my comments for details. Further, I think the u\*-threshold estimation and removal of data with low u\* values is nothing to debate about as the eddy covariance technique is not working under those conditions and this must be accounted for in the data processing. For further details see the attached pdf:*

   *I would highly recommend a standardized data processing for the gap filling and flux partitioning. The REddyProc (Wutzler et al., 2018) package is easy to use and has all needed functions to do the u\*-threshold estimation, subsequent gap-filling, and finally the flux partitioning of NEE into Reco and GPP. I think the argument that in this specific case the standardized methods are not working does not hold. In this context I would like to question the QC scheme used here. All fluxes between QC 1 and QC 8 (based on the Foken 1-9 system) are used but here we are already quite far from the assumptions which allow us to apply the eddy covariance technique. I would suggest to go for QC 1-6 to maintain reliable data.*

The authors are surprised by the main statement of the reviewer, that below a $u_*$-threshold, e.g. 0.28 m s$^{-1}$ (Wutzler et al., 2018) the eddy-covariance method cannot be applied anymore. This would mean that fundamental works on turbulence in the near surface layer, see e.g. Businger et al. (1971) Fig. 8, would have to be revised and all universal functions, especially in the stable range, would be faulty, because in the opinion of the reviewer, the eddy-covariance method can no longer be applied under these conditions. Rather, the pioneers of atmospheric turbulence and the eddy-covariance method have investigated similarity relations that clearly describe the existence of a well-developed turbulence (Wyngaard et al., 1971;Panofsky and Dutton, 1984;Monin and Obukhov, 1954). These have been used to detect a developed turbulence for which the eddy-covariance method can be applied (Foken and Wichura, 1996;Vickers and Mahrt, 1997;Foken et al., 2017).

It is true, however, that by using a $u_*$-threshold one can be sure that the measurements are not influenced by intermittency, decoupling, gravity waves, low level jets etc. and that no further investigations are necessary. For large measurement programs such as ICOS with central data processing, such a method makes sense, since models can be used in addition to the classical gap-filling methods. However, the present study was not created in the context of such a program, but is a process study, where it is important to include in the study as much original data as possible with developed turbulence and to limit the gap-filling to a necessary minimum because of the reported difficulties to parameterize the equations used for gap-filing. Fig. 1 shows that at a $u_*$ threshold of 0.28 m s$^{-1}$ 37 % data can still be excluded. Together with the 10% of gap-filled data (see Supplement Figure S8), almost half of all data would have to be replaced

by parameterized data. This is certainly possible for long-term studies, but not for a process study. This would mainly exclude unstable and stable measurements showing particularly high or low fluxes compared to simple model assumptions (Fig. 2). This would also mean that all weak wind situations below about 2 m s$^{-1}$ are not considered (Fig. 3). The authors therefore decided not to perform the data processing according to a routine method, but to use all possibilities offered by the relevant publications (Foken et al., 2004;Foken et al., 2012a;Foken et al., 2012b;Mauder et al., 2013). Critical is that these are many self-quotes, but the work has been developed together with renowned scientists from this field, such as L. Mahrt, R. Leuning and M. Aubinet.

[Figure]

Fig. 1: Frequency of the friction velocity

[Figure]

Fig. 2: Friction velocity as a function of stability (z/L, z: height, L: Obukhov length)

[Figure]

Fig. 3: Dependence of the friction velocity on the wind speed

The answers to the following major and minor comments of the reviewer must be considered in the light of these statements, so that details are not repeated again. It is sometimes difficult to separate the results and discussion sections for reasons of readability and understanding. The authors will therefore change the headings:

R**esults → Results and specific discussions;**

**Discussion → Overall discussion**

> *2. .Within the manuscript for different analysis different QC schemes are used which is very confusing. Sometimes wind sectors are removed then again, they are not. In one instance QC 1-6 is used in another QC 1-9 this is confusing for the reader because it jumps back and forth.*

The authors do not use two different QC schemes. We only use QC 1-6 for gap-filling (Supplement S7), otherwise Q 1-8 (Supplement S8) if we carefully check the causes of the bad flags. **The authors will check the text again to see if this is clear enough.**

The background is the recommendation that for basic investigations such as the determination of the functions for gap-filling only classes 1-3, if possible 1-6 should be used (see Foken et al., 2012b, p. 117-118). This option is only available for individual data processing.

> *3. The uptake in burned area after the fire is either a measurement artefact or it comes from vegetation that is still taking up carbon. Maybe arising from imperfect footprint estimates? Or a too tall canopy height which will reduce the footprint size and thus suggest that footprints are smaller than they are in real (minor comments below)? Which in turn would reduce the number of footprints having 80% of Maritime Pine contribution. In the east there is the eucalyptus patch close to the tower. Not much is mentioned about it but it seems to be the second most important wind direction and thus potentially influencing the measurements?*

The authors actually already referred these hypotheses on lines 522-525: "Possibly, this immediate post-fire photosynthetic activity originated from various patches of pines with scorched crowns to the northeast and south of the EC tower and/or from re-sprouting eucalypts, in particular the 4 individual trees near the tower and/or the patch to the east of it."

Elaborating further on this matter, the authors would like to clarify that the patches of scorched pine crowns were present between 22 September 2017 and 03 January 2018 but no longer on 05 March 2018 (as the pine needles had dropped to the forest floor). This can be seen on the ortho-photomaps included at the bottom of this rebuttal. The eucalypt patch to the east of the tower (see Figure 1 in the Manuscript) could be an alternative or complementary source, either through post-fire photosynthetic activity of the scorched crowns (visible – as shadows - on the 22 Semptember2017 ortho-photomap, at least for the closest trees) or through newly emerging sprouts. However, the relevance of this potential eastern source is probably reduced, due to the low frequency of the eastern sector (see Figure 2 of the Manuscript and answer to minor comment 16). The 4 individual eucalypt trees at relatively close distance to the west of the tower, shown in Figure 1 of the Manuscript, could also be an alternative or complementary source. However, all 4 trees suffered complete crown consumption by the fire, as is visible on the 22 September 2017 ortho-photomap, and only showed very reduced recovery three months later, as is evident on the 03 January 2018 ortho-photomap. In resume, the authors believe that the scorched pine patches are the most likely source of the observed carbon assimilation immediately after the fire but we have not measured the photosynthetic activity of scorched pine needles and are also not aware of any previous study on that.

Since the burnt area around the tower mainly consist of maritime pine woodland, a smaller footprint is not expected to reduce the number of footprints with 80% pine

Although the text explicitly refers that the fire burnt a total area of 12.5 $km^2$ (line 65), the map can indeed be somewhat misleading as to the extent of the burnt area around the study site. The minimal distance to the unburned area was 1.1 km in the W-NW sector. This is outside the footprint.

It is possible that the text does not make it clear enough that the forest fire was a large-scale fire, i.e. the areas adjacent to the area under investigation were also completely burned down. While in the case of pine trees only single charred tree trunks remain, eucalyptus trees still have completely dried leaves, but without any assimilation or respiration function. **The authors will clarify this again in the text.**

The authors also investigated the question of possible artifacts. We have selected two periods with nearly equal energy input for which > 80% of the footprint is determined by pine (Fig. 10) and eucalyptus (Fig. S12), respectively. There were no significant differences in the $CO_2$ fluxes, so that a "not exact" footprint does not lead to artifacts.

The question of eucalyptus trees in the east is examined in detail in section 3.1.1. The sonic anemometer type CSAT3 was set up in such a way that the impermissible backward flow is largely identical with the flow through the trees. As Fig. 6 shows, this leads to a reduction of the data quality. In some cases, the data quality is so bad due to mechanical turbulence that the data were replaced. Looking at the investigations in sections 3.1.1 and 3.3.3 it becomes clear that these trees only have an influence on the mechanical turbulence and not on the $CO_2$ flow.

4. *The data set associated to this discussion paper is only containing daily data for NEE, GPP and Reco. But most analysis are based on half hourly data. I guess it would be nice to also have those data available in the dataset. But this depends on the journal / editors decision.*

As far as the authors know, in long-term studies (e.g. Keenan et al., 2014) are used as common units g C $m^{-2}$ $d^{-1}$ or g C $m^{-2}$ $yr^{-1}$. Thus, a user of the data no longer needs to convert µmol $m^{-2}$ $s^{-1}$. Since we are still exploring further the half hourly data set for further studies, we have opted for only making available the daily data set, at least for the time being. At the same time, lines 456-457 explicitly state that "Other data can be requested by email to bruna.oliveira@ua.pt."

**Minor comments**

1. L42-43 not fully clear if the ecosystem is still a source on an annual basis or considering the totally emitted carbon vs. the total carbon sequestered by the ecosystem during the following 10 years.

The reference Dore *et al.* (2008) is included in Table S1. In summary, the authors only measured the ecosystem fluxes 10-years after a stand-replacing wildfire. Even though the authors write "Ten years after the fire, the burned site was still a source of $CO_2$ to the atmosphere (…)" they do not present data that support a constant behavior during the 10 years. Hence, the ecosystem might have behaved as a C sink in previous years.

**Following the comment of the reviewer, the text in the Manuscript was slightly adapted from**

*"Furthermore, these effects can be long lasting, as well illustrated by Dore et al. (2008), finding that a 10-year old burnt site was still a carbon source."*

**to**

*"Furthermore, these effects can be long lasting, as illustrated by Dore et al. (2008), finding that a Pinus ponderosa forest was a carbon source 10-years after a stand-replacing wildfire (Supplementary Material Table 1)."*

2. L57 maybe adding a city or village would make it easier to find.

Following the suggestion of the reviewer, the village "Vila de Rei" was added to the text. Since the installation is still in operation and the equipment is without fencing or other type of protection, the authors do not want to disclose the exact location of the study area to avoid vandalism, as experienced in the past by the team.

The original sentence was:

"*Median height and diameter-at-breast-height of the burnt pine trees ranged from 4.6 to 6.7 m and from 2.5 to 4.3 cm, respectively.*"

**Following the comment of the reviewer, the sentence was updated to:**

*"Median height and diameter-at-breast-height of the burnt pine trees in each of the 5 plots selected for fire severity assessment ranged from 4.6 to 6.7 m and from 2.5 to 4.3 cm, respectively."*

**to make more clear that 4.5 – 6.7 is not the median tree height in the footprint area and refers only to the fire severity assessment plots.**

4. Figure 1 Include the slope angle map from the supplementary figure 1 and also add a map with the footprint climatology. Also add the points for the fire severity sampling into the map.

**Following the suggestion of the reviewer, the slope angle map will be transferred from the supplementary material to the manuscript and the fire severity assessment points will be added to one map in the Supplementary material.**

The authors do not have a Footprint climatology (Amiro, 1998;Göckede et al., 2008), because the present study is not a long-term study but a process study. We do not see the cumulative flux as the main result of the present paper. The reported wind climatology and footprint distribution allow sufficient conclusions about the footprint climatology.

*5. L100 Based on what was the zero-plane displacement set to 3.8m and the canopy height to 8m if the median height only goes up to 6.7 m? This seems wrong.*

The authors agree that the canopy height of 8 m may lead to some confusion, given the information provided on tree height in the 5 fire severity assessment points in L78-79. As mentioned in a previous comment, **the text in L78-79 was reformulated** to:

*"Median height and diameter-at-breast-height of the burnt pine trees in each of the 5 plots selected for fire severity assessment ranged from 4.6 to 6.7 m and from 2.5 to 4.3 cm, respectively."*

In L98-99 the authors wrote:

*"The – standing - pine trunks in the immediate surroundings of the tower were approximately 8 m high, and this was used as "canopy" height in all calculations, together with a zero-plane displacement of 3.8 m."*

Hence, the authors believe that with the reformulation in L78-79 the information on canopy height is now more clear.

The determination of zero-plane displacement proves to be extremely complicated in a heterogeneous area, especially if there are only single roughness elements (charred trees). We followed the recommendation in this regard and evaluated only the highest roughness elements (Leclerc and Foken, 2014, p. 34). An exact determination would only be possible by parallel profile and flux measurements (Foken, 2017). Since areas with burned pine trees and eucalyptus in the $CO_2$ fluxes do not differ significantly, the error on the footprint determination should be small.

*6. Table 1 I assume the wind vector components at 20Hz were likewise sampled with the CSAT 3 correct? If so please insert that for "Sensor" instead of sonic anemometer.*

**The authors updated Table 1** to make more clear that the sensor CSAT3 was used to determine the wind vector (3 components) and sonic temperature.

*7. Table 1 The EC5 Sensors are not mentioned in the table but should be included.*

**Following the suggestion of the reviewer, the EC5 sensors were added to Table 1**:

| Height | Parameter | Sensor | Manufacturer | Remarks |
|---|---|---|---|---|
| (…) | | | | |
| -2.5; -7.5; -10; -20; -30 cm | Soil temperature and volumetric water content | GS3 sensor linked to a Em50 data logger | GS3 sensor linked to a Em50 data logger | 1.5 m from tower; 0.02 Hz data |
| -2.5; -7.5 cm | Soil volumetric water content | EC5 sensors linked to Em50 data loggers | DECAGON Devices | 3 EC5 at -2.5 cm and 1 EC5 at -7.5 cm in each of 5 points along the footprint area |
| -2.5 cm | Soil temperature and volumetric water content | GS3 sensors linked to Em50 data loggers | DECAGON Devices | 1 GS3 in 5 points along the footprint area |

and the corresponding text in L104-109 was clarified, also considering minor comments 14 and 15 by the reviewer:

*"In addition to the soil moisture/temperature station immediately next to the tower listed in Table 1, five soil moisture/temperature stations were installed along the above-mentioned transect. Each station comprised 4 EC5 soil moisture sensors and 1 GS3 soil moisture/temperature. Of the sensors in each of the 5 stations, 1 EC5 was near the roots of a burnt pine tree, 1 EC5 was near the roots of a reprouter shrub (Pterospartum tridentatum), 1 EC5 at -2.5 cm and 1 EC5 at -7.5 cm in the inter-patch (clearing), and 1 GS3 at -2.5 cm in the inter-path. In this study, only the data from the soil moisture sensors installed at -2.5 cm at the inter-patches were used."*

8. *L129 – 131 What percentage of the half hourly footprint contribution were used to determine the contribution of the pine trees (e.g. 80%, 90%, 95%)?*

The distribution of pine trees in the Footprint sector is shown in Figure 4. For the process studies (non-cumulative flux) only data with Footprint > 80% were used. Note: Footprint models determine the Footprint area (effect level) relatively accurately, but the exact local allocation is often insufficient due to the influence of crosswind fluctuations (Markkanen et al., 2009, 2010).

9. *L138 the supplementary material is not ordered as they appear in the text. This should be Figure S2. Please change accordingly throughout the manuscript and supplementary material accordingly.*

The authors appreciate that the reviewer pointed this. **The order of the Figures in Manuscript and Supplementary Material will be revised.**

10. *L150-152 The night time estimation is very coarse and might create huge biases especially during the winter period when sunrise and sunset are clearly later and earlier, respectively. I would recommend to use the potential radiation to estimate day- and nighttime. This is for example included in the REddyProc package that can also take care of the gap-filling as well as flux partitioning including u\*-threshold estimation.*

The gap-filling of the data was done according to the exact sunrise and sunset times (Supplement Figure S8). A significantly tighter time frame was chosen for the processing of the parameterization of the data for gap-filling. Thus, data with significant sensible heat flows were excluded in order to keep the additive WPL correction, which is in the order of magnitude of the flux, as low as possible (Webb et al., 1980, Fig. 1).

**The sentence:**

L278 *"Respiration fluxes were determined using only the measurements from 10 pm to 04 am UTC, with more than 80% of the footprint area from the Maritime Pine stands, and with quality flags 1-6."*

**will be reformulated to:**

*"Respiration fluxes were determined using only the measurements from 10 pm to 04 am UTC* **to reduce the influence of the additive WPL correction (Webb et al., 1980)***, with more than 80% of the footprint area from the Maritime Pine stands, and with quality flags 1-6."*

11. *L157 I think "initial slope" is better than "linear slope" given that the function is not linear.*

**Following the comment of the reviewer, the text will be updated.**

12. *L158-159 I would argue that α and Qc,sat are no constant but variable based on environmental conditions which are not just a function of air temperature but also of vapor pressure deficit (VPD), soil water content (SWC) and status of the vegetation. Qc,sat will increase based on the leaf area index (LAI). Was this procedure done for individual periods or windows e.g. 4 weeks? This would be important to clarify.*

The procedure was done for temperature classes only, and the results are summarized in Table 3. **The authors will include "for temperature classes" in the table.** LAI determination was not yet possible in the first year after the fire (see Supplementary Figures p. 14)

*13. L168-169 what were the upper and lower threshold for the mean vertical wind speed?*

Following the comment of the reviewer, the authors noticed that the 1-minute input files for the wavelet analysis were coordinate rotated. **The authors will change the text accordingly.**

*14. L177 Why was not 10 and 20cm used and instead the average? Further your first sensor is at -2.5 cm so the integral should be from -0.025 to -0.15 (equation 3). From Table one it seems that only the sensors 1.5. to the tower were used but not the whole transect. Is this correct? If so I would suggest to also use the other profiles from the transect to get a better spatially representative average soil heat flux.*

Equation 3 is in line with the references Liebethal and Foken (2007) and Yang and Wang (2008). We have therefore omitted further details. The data of the temperature sensors above 15 cm depth have of course been included in the determination of the storage term ($2^{nd}$ term of the equation). The depth 10 and 20 cm were needed to determine the heat flux at 15 cm depth. The soil sensors along the transect did not measure deeper than 7.5 cm, as now listed in Table 1 and explained in L104 (please see answer to minor comment 7). Data from these sensors was only used to determine the volumetric soil water content classes, as detailed in Section 2.3.5.

*15. L186 not clear what inter-patch means. I assume the sensors from the transect. Please indicate this also in table 1 and in lines 107-108. Which sensors were used for what? It is not consistent because the deeper VWC sensors were also used for estimating the soil heat flux.*

Following this and related comments of the reviewer (minor comments 7 and 14), **the text in L104-109 was updated** to make the experimental setup more clear:

*"In addition to the soil moisture/temperature station immediately next to the tower listed in Table 1, five soil moisture/temperature stations were installed along the above-mentioned transect. Each station comprised 4 EC5 soil moisture sensors and 1 GS3 soil moisture/temperature. Of the sensors in each of the 5 stations, 1 EC5 was near the roots of a burnt pine tree, 1 EC5 was near the roots of a reprouter shrub (Pterospartum tridentatum), 1 EC5 at -2.5 cm and 1 EC5 at -7.5 cm in the inter-patch (clearing), and 1 GS3 at -2.5 cm in the inter-path. In this study, only the data from the soil moisture sensors installed at -2.5 cm at the inter-patches were used."*

*16. L209 what is the value of q for your data set?*

The value for the q parameter in Equation 5 used in each MAD analysis is referred in the Results section: for the Energy balance closure, q = 0.5 was used (L261); and for the gap-filling of respiration, q= 0.5 was used (L81).

*17. L230-231 If all data from that wind direction (30 – 90 degree) are flagged and removed this should be included in the schemes for data treatment. Also, it would be helpful if Figure 2 would be adjusted in a way that the 30° sectors fit to the bins of figure 6.*

Please see major comment 3: This comparison is not possible. Figure 2 are mean data. Fluxes are based on turbulence data with a significant variation of the cross-wind component, which can be up to ± 45° for the wind direction (see e.g. Foken and Leclerc, 2004, Fig. 7). Besides the roughness influence there is a significant stability influence (Blackadar, 1997). Thus, the data selection was made according to the individual quality classes and not generally according to the mean wind sector.

18. *Figure S8 more details on the individual symbols and numbers could be needed. I think total number of removed data is quite low given all quality criteria mentioned. If I see that correctly there are about 10-15 % of all data missing to their wind direction (30 – 90°), further there are 40 % of data with not enough Maritime Pine coverage in the footprint area. But from Figure S8 it looks like only 1727 data points were removed and gap filled.*

Please see major comment 3 and minor comment 17.

19. *Figure S4 the different grey colors are hard to distinguish. Would be nice to add different symbols to the individual lines. Additionally, this figure shows nicely how wrong a fixed period between 10pm and 4 am is for night time estimates. Even around end of June the sunrise is just around 6am UTC when considering the increase in H and net radiation. The same is true for the nighttime.*

**The authors will add symbols to the Figure.** For the second part, please see minor comment 10.

20. *L245-248 What about heat storage in the burned trunks of the trees? This is something that is not considered or discussed here. The half hourly fluxes are missing the storage term of heat but this is not discussed. Additionally, the trunks certainly contribute to the sensible heat fluxes and increase the surface area compared to the bare soil. Further it is not clear if these trees are also fully in the field of view of the net radiometer. Especially in the morning at high solar zenith angle the trees will absorb a lot of the short-wave incoming radiation, heat up and contribute to the sensible heat flux. This would happen in the morning and the afternoon. Something for the discussion I guess.*

The present study is not to investigate the closure of the energy balance. For this purpose, all conditions are missing both with regard to instrumentation and necessary additional information (Mauder et al., 2020). For this reason, the figures are not in the article but in the supplement for interested readers. Such inadequacies are the choice of the net-radiometer, the heterogeneity of the surface with soil, ashes, wood residues etc., and also the still standing charred tree trunks (the number is not large, because in Figure 3 you can still see the horizon well). The examination of the energy balance is in the sense of quality control (Foken et al., 2004;Foken et al., 2012b) and a proper execution of the measurements can be assumed with a closure of 80-100 %. The authors do not see the need to complete the text.

21. *Figure S5 The linear equation is very small and hard to read including the r² values. As you did not provide an offset value in the formula I assume you forced the intercept to be 0? Please state that explicitly if it is the case. If the detection limit of 10 W m-2 was used then this should be an absolute value. During nights with wind you might get negative sensible heat fluxes which can be larger than -10 and the same is true for latent heat during those nights just that it will in most cases not be negative. Also mention that there is the confidence interval of the linear fit included as the grey shaded area. The number of points going into the linear relation estimate should also be mentioned.*

**The authors will improve the visual quality of the Figures**. Due to the reasons for the closure of the energy balance, night time measurements should be excluded from the assessment of the residual, which the authors have done. The authors will not go into this further here, as this is comprehensively stated in review articles and the references cited there (Foken, 2008;Mauder et al., 2020).

22. *L257-259 how do we know that net radiation was underestimated? Due to the tilting? Is there a hysteresis in the net radiation and we do know that there is overestimation in the afternoon and underestimation in the morning caused by the tilting toward SW (L253)? This is not clear at all. This tilting would overestimate the incoming components in the afternoon and underestimate them in the morning which could then lead to the observed pattern. Is that the reasoning? How do we know there is an overestimation in the long wave outgoing? What evidence do we have for that? Many of these thoughts belong in the discussion and not the results I would say.*

It is not the tilting because, as said in the comment, this leads to both over- and under-determination. All net-radiometers that do not measure all 4 radiation components separately have the problem of too low readings (10-20%). This has basically been investigated by Halldin et al. (1992). This is also discussed in the overview papers on energy balance (Foken, 2008). The authors have quoted a paper specifically on the problems of NR-lite (Brotzge and Duchon, 2000) and believe that in Lines 253-257 this is sufficiently stated.

L235: *"Net radiometers, as the one used in this study (Table 1), that do not measure the four up- and down-welling long- and shortwave radiation components separately, are well-known to underestimate net radiation (Kohsiek et al., 2007)."*

23. *L264-270 this is all discussion and no results. And even for the discussion I think this is not really needed. Your manuscript is not about the problems of energy balance closure under post-fire conditions but to characterize ecosystem fluxes. I find this whole hypothetical discussion on where the non-closure is quite distracting from the main objective. And we all know that the energy balance is not closed at any site. With you lack of 10% you are actually closer than most sites around the globe.*

The authors agree with this comment, please see answer to minor comment 22. However, it is important to determine whether the $CO_2$ flux must be corrected or not.

24. *L273-276 I don't agree at all with statement. I don't see why the development of vegetation is a reason not to use standard gap-filling procedure? There are many grasslands in the Mediterranean or other dry areas which senesce during the summer and regreen during the autumn and winter. It is the same as happening here. And all of these sites in FLUXNET use the standard gap filling and flux partitioning schemes similar to the one used in REddyProc. So this is not a valid reason. The second point is somewhat true because moving windows smear the signal when rapid changes are happening e.g. the re-wetting of the ecosystem after the summer. Still if you are interested in the annual sums which also seems to be a point of the manuscript I would highly suggest to use standard gap-filling and flux partitioning.*

In the authors opinion, it is doubtful whether when can directly compare seasonal patterns in plant activity in a Mediterranean and dryland grasslands with vegetation recovery from wildfire. The present results also demonstrate this, in the sense that they show basically no vegetation activity during the start of the autumn rainfall season as would be expected from a system reviving from summer senescence.

According to Papale (2012) there are different methods to achieve an optimal gap-filling. The method of Wutzler et al. (2018) is one but not the only one. The authors have followed the basics of gap-filling (Falge et al., 2001a, b) to the same extent as it was done in all other papers. The only difference is that we did not use a method with a $u_*$-criterion. We have explained this extensively in major comment 1. Instead, the authors used the method of Ruppert et al. (2006), in which the comparison to the $u_*$-method was also made.

It must be added that the fluxes in the present work are very small and that, in addition to the usual processes of assimilation and respiration, the microbial decomposition of ashes also plays a role on which there is little knowledge so far.

25. *L289 E0 "0" must be subscript*

**Following the comment of the reviewer, the text was updated.**

26. *L300-305 Formula 6 is not fully explained. What is T in this formula? Is it the surface temperature as assumed by the Stefan-Boltzman law? How was this then estimated or is it the air temperature. The reasoning and how this was derived should be explained in the methods and not in the results. I would recommend to move this part to methods section. And at the end of line 305 a point is missing.*

The authors apologize for missing half sentence stating what T in Equation 6 is.

L305 was updated to *"where $\sigma_{SB}$ is the Stefan-Boltzmann constant and T is the temperature at 11.8m height. "*

This very rough estimation is possible considering error considerations for long wavelength radiation (Gilgen et al., 1994). It is certainly useful to move this to the methods section. However, there are only 4 lines and it could affect the clarity. The authors will decide this after presentation of the other reviews.

27. *3.2.3."Generating the final data set" this part is not consistent with what is written in the methods and what is shown in the schematics. Here night time is defined different as compared to the methods part (L150). Here nothing is mentioned about the filter of wind direction or the footprint filter. This is confusing.*

For time please see minor comment 9. **The authors will change the section title to "3.2.3 Generating the final data set for cumulative fluxes"**. For the process studies, as noted there, for example, higher demands were made on the footprint.

28. *L336 first time that the storage term is mentioned. This should be made clear in the methods part. And it should be mentioned if it was the 1-point storage correction or based from a multiple point vertical profile system.*

Following the reviewer suggestion, **it was added to the Methods (L126) that the software packaged used, TK3, estimates the $CO_2$ storage flux from one point $CO_2$ measurements** as suggested by Hollinger et al. (1994).

29. *Figures S9 and S10 it is very hard to associate the events across variable of the plots. First the horizontal zero lines should be included. Secondly the areas with dew should be shaded vertically so the points can be clearly associated and compared across the different variables. To me it looks like we see a flushing of CO2 in the morning that accumulated during the night at the ground. If the 1-point storage correction was used this one will for sure not capture what is happening at the ground. How was the friction velocity during the night? This could also tell a bit about the storage of CO2 at the ground. If the moment of the high CO2 flux goes together with an increase in u\* then we can be quite sure that this was rather a flush of CO2 than CO2 originating from instantaneous microbial activity or water flowing into pores and flushing out CO2. Also the argument of the negative sensible heat flux is not very convincing. Dew evaporation needs energy but we must separate the individual processes. As long as we have a negative net radiation and we reached already 100% relative humidity we will observe more dewfall which means a negative latent heat flux (this was actually not shown in figure 8). Only when energy is provided in form of incoming short-wave radiation we can have conditions of evaporation. But then the energy for evaporation is coming from the incoming short-wave radiation which means sensible heat is not increasing fast but slow. The negative sensible heat flux looks to me like an inversion of cold air at the surface was breaking up in the morning and transporting cold air upwards together with the CO2. I think here we are missing many data streams to bring the story together.*

The authors have investigated this case very carefully: Figures S9 and S10 were included in the supplement to show the relevant conditions on the days from September 26 to 29. The discussion of the dew fall can only be seen in Figure 8. The friction velocity at the relevant time was 0.05-0.10 m s$^{-1}$, so that a dynamic effect can be excluded. Also the assumption of the influence of the storage term is not applicable, because the fluxes were calculated with the wavelet tool (Schaller et al., 2017). The sensible heat flux is a good indicator for the dewfall, because then it is positive, while it is negative for evaporation without additional short wave radiation. The latent heat flux is too small to represent the process well.

The statement that short-wave radiation is necessary for evaporation (evaporation, not transpiration) is wrong. The example shown here can be classified as an oasis effect (Stull, 1988). An impressive example is shown in Foken (2017, Fig. 1.14). Evaporation occurs when moisture is present, a water vapor pressure gradient exists, and turbulence is present. If no short wave

radiation is present, the energy can also be provided by long wave cooling (negative sensible heat flux) or the ground heat flux.

**The authors will improve the visual quality of the Figures.**

30. *Figure S12 and S13 here it seems very clear that the burned area is more heterogeneous as expected. Else the fluxes would be more homogeneous and not scatter so much. This indicates that there are patches which take up carbon. Maybe a better representation would be a plot of midday NEE or CO2 flux vs wind direction. I think this would make it clearer and highlight the differences between the burned and the other areas and maybe also give you more trust in the differences you see. Maybe I would even do it for every month/ season but this is just a suggestion.*

The dry cases Figures 10 (pine) and S12 (eucalyptus), which were selected for the respective surface with > 80% in the footprint regarding all 30-minute fluxes, must be compared. No significant fluctuations can be detected. The magnitude of the fluctuations is typical for the influence of turbulence, low cloud cover etc.  Figure S13 shows a case after/with precipitation (also footprint > 80% for pine). Here the fluctuations are naturally larger, whereby surely different humidification and cloudiness play a role. After the wildfire there were no unburnt areas in the close vicinity of the tower and it was only in April 2018 that the green areas became visible (Fig. 04).

See also answer to major comment 3.

[Figure]

03 January2018

50    0    50    100    150    200 m

05 March 2018

[Figure]

100    0    100    200    300    400 m

Fig. 04: UAV-based ortho-photomaps of the burnt area to the west of the EC tower (marked as a red triangle) on 22 September 2017 and 03 January 2018, and of the burnt area surrounding the EC tower on 05 March 2018.

**References**

Amiro, B. D.: Footprint climatologies for evapotranspiration in a boreal catchment, Agric. For. Meteorol., 90, 195-201, doi, 1998.

Blackadar, A. K.: Turbulence and Diffusion in the Atmosphere, Springer, Berlin, Heidelberg, 185 pp., doi, 1997.

Brotzge, J. A., and Duchon, C. E.: A Field Comparison among a Domeless Net Radiometer, Two Four-Component Net Radiometers, and a Domed Net Radiometer, J. Atm. Oceanic Techn., 17, 1569-1582, doi: 10.1175/1520-0426(2000)017<1569:AFCAAD>2.0.CO;2, 2000.

Businger, J. A., Wyngaard, J. C., Izumi, Y., and Bradley, E. F.: Flux-profile relationships in the atmospheric surface layer, J. Atmos. Sci., 28, 181-189, doi: 10.1175/1520-0469(1971)028<0181:FPRITA>2.0.CO;2, 1971.

Falge, E., Baldocchi, D., Olson, R., Anthoni, P., Aubinet, M., Bernhofer, C., Burba, G., Ceulemans, R., Clement, R., Dolman, H., Granier, A., Gross, P., Grunwald, T., Hollinger, D., Jensen, N. O., Katul, G., Keronen, P., Kowalski, A., Lai, C. T., Law, B. E., Meyers, T., Moncrieff, H., Moors, E., Munger, J. W., Pilegaard, K., Rannik, U., Rebmann, C., Suyker, A., Tenhunen, J., Tu, K., Verma, S., Vesala, T., Wilson, K., and Wofsy, S.: Gap filling strategies for defensible annual sums of net ecosystem exchange, Agric. For. Meteorol., 107, 43-69, doi, 2001a.

Falge, E., Baldocchi, D., Olson, R., Anthoni, P., Aubinet, M., Bernhofer, C., Burba, G., Ceulemans, R., Clement, R., Dolman, H., Granier, A., Gross, P., Grunwald, T., Hollinger, D., Jensen, N. O., Katul, G., Keronen, P., Kowalski, A., Lai, C. T., Law, B. E., Meyers, T., Moncrieff, H., Moors, E., Munger, J. W., Pilegaard, K., Rannik, U., Rebmann, C., Suyker, A., Tenhunen, J., Tu, K., Verma, S., Vesala, T., Wilson, K., and Wofsy, S.: Gap filling strategies for long term energy flux data sets, Agric. For. Meteorol., 107, 71-77, doi, 2001b.

Foken, T., and Wichura, B.: Tools for quality assessment of surface-based flux measurements, Agric. For. Meteorol., 78, 83-105, doi: 10.1016/0168-1923(95)02248-1, 1996.

Foken, T., Göckede, M., Mauder, M., Mahrt, L., Amiro, B. D., and Munger, J. W.: Post-field data quality control, in: Handbook of Micrometeorology: A Guide for Surface Flux Measurement and Analysis, edited by: Lee, X., Massman, W. J., and Law, B., Kluwer, Dordrecht, 181-208, doi, 2004.

Foken, T., and Leclerc, M. Y.: Methods and limitations in validation of footprint models, Agric. For. Meteorol., 127, 223-234, doi, 2004.

Foken, T.: The energy balance closure problem – An overview, Ecolog. Appl., 18, 1351-1367, doi: 10.1890/06-0922.1, 2008.

Foken, T., Aubinet, M., and Leuning, R.: The eddy-covarianced method, in: Eddy Covariance: A Practical Guide to Measurement and Data Analysis, edited by: Aubinet, M., Vesala, T., and Papale, D., Springer, Dordrecht, Heidelberg, London, New York, 1-19, doi: 10.1007/978-94-007-2351-1_1, 2012a.

Foken, T., Leuning, R., Oncley, S. P., Mauder, M., and Aubinet, M.: Corrections and data quality in: Eddy Covariance: A Practical Guide to Measurement and Data Analysis, edited by: Aubinet, M., Vesala, T., and Papale, D., Springer, Dordrecht, Heidelberg, London, New York, 85-131, doi: 10.1007/978-94-007-2351-1_4, 2012b.

Foken, T.: Micrometeorology, 2$^{nd}$ ed., Springer, Berlin, Heidelberg, 362 pp., doi: 10.1007/978-3-642-25440-6, 2017.

Foken, T., Göckede, M., Lüers, J., Siebicke, L., Rebmann, C., Ruppert, J., and Thomas, C. K.: Development of flux data quality tools, in: Energy and Matter Fluxes of a Spruce Forest Ecosystem, Ecological Studies Vol. 229, edited by: Foken, T., Springer, Cham, 277-308, doi: 10.1007/978-3-319-49389-3_12, 2017.

Gilgen, H., Whitlock, C. H., Koch, F., Müller, G., Ohmura, A., Steiger, D., and Wheeler, R.: Technical plan for BSRN data management, WRMC, Techn. Rep., 1, 56 pp, doi, 1994.

Göckede, M., Foken, T., Aubinet, M., Aurela, M., Banza, J., Bernhofer, C., Bonnefond, J.-M., Brunet, Y., Carrara, A., Clement, R., Dellwik, E., Elbers, J. A., Eugster, W., Fuhrer, J., Granier, A., Grünwald, T., Heinesch, B., Janssens, I. A., Knohl, A., Koeble, R., Laurila, T., Longdoz, B., Manca, G., Marek, M., Markkanen, T., Mateus, J., Matteucci, G., Mauder, M., Migliavacca, M., Minerbi, S., Moncrieff, J. B., Montagnani, L., Moors, E., Ourcival, J.-M., Papale, D., Pereira, J., Pilegaard, K., Pita, G., Rambal, S., Rebmann, C., Rodrigues, A., Rotenberg, E., Sanz, M. J., Sedlak, P., Seufert, G., Siebicke, L., Soussana, J. F., Valentini, R., Vesala, T., Verbeeck, H., and Yakir, D.: Quality control of CarboEurope flux data – Part 1: Coupling footprint analyses with flux data quality assessment to evaluate sites in forest ecosystems, Biogeosci., 5, 433-450, doi: 10.5194/bg-5-433-2008, 2008.

Halldin, S., and Lindroth, A.: Errors in net radiometry, comparison and evaluation of six radiometer designs, J. Atm. Oceanic Techn., 9, 762-783, doi, 1992.

Hollinger, D. Y., Kelliher, F. M., Byers, J. N., Hunt, J. E., McSeveny, T. M., and Weir, P. L.: Carbon dioxide exchange between an undisturbed old-growth temperate forest and the atmosphere, Ecology, 75, 134-150, doi, 1994.

Keenan, T. F., Gray, J., Friedl, M. A., Toomey, M., Bohrer, G., Hollinger, D. Y., Munger, J. W., O/'Keefe, J., Schmid, H. P., Wing, I. S., Yang, B., and Richardson, A. D.: Net carbon uptake has increased through warming-induced changes in temperate forest phenology, Nature Clim. Change, 4, 598-604, doi: 10.1038/nclimate2253, 2014.

Leclerc, M. Y., and Foken, T.: Footprints in Micrometeorology and Ecology, Springer, Heidelberg, New York, Dordrecht, London, XIX, 239 pp., doi: 10.1007/978-3-642-54545-0, 2014.

Liebethal, C., and Foken, T.: Evaluation of six parameterization approaches for the ground heat flux, Theor. Appl. Climat., 88, 43-56, doi: 10.1007/s00704-005-0234-0, 2007.

Markkanen, T., Steinfeld, G., Kljun, N., Raasch, S., and Foken, T.: Comparison of conventional Lagrangian stochastic footprint models against LES driven footprint estimates, Atmos. Chem. Phys., 9, 5575-5586, doi: 10.5194/acp-9-5575-2009, 2009.

Markkanen, T., Steinfeld, G., Kljun, N., Raasch, S., and Foken, T.: A numerical case study on footprint model performance under inhomogeneous flow conditions, Meteorol. Z., 19, 539-547, doi, 2010.

Mauder, M., Cuntz, M., Drüe, C., Graf, A., Rebmann, C., Schmid, H. P., Schmidt, M., and Steinbrecher, R.: A strategy for quality and uncertainty assessment of long-term eddycovariance measurements, Agric. For. Meteorol., 169, 122-135, doi: 10.1016/j.agrformet.2012.09.006, 2013.

Mauder, M., Foken, T., and Cuxart, J.: Surface Energy Balance Closure over Land: A Review, Boundary-Layer Meteorol., online first, doi: 10.1007/s10546-020-00529-6, 2020.

Monin, A. S., and Obukhov, A. M.: Osnovnye zakonomernosti turbulentnogo peremesivanija v prizemnom sloe atmosfery (Basic laws of turbulent mixing in the atmosphere near the ground), Trudy geofiz. inst. AN SSSR, 24 (151), 163-187, doi, 1954.

Panofsky, H. A., and Dutton, J. A.: Atmospheric Turbulence - Models and Methods for Engineering Applications, John Wiley and Sons, New York, 397 pp., doi, 1984.

Papale, D.: Data gap filling, in: Eddy Covariance: A Practical Guide to Measurement and Data Analysis., edited by: Aubinet, M., Vesala, T., and Papale, D., Springer, Dordrecht, Heidelberg, London, New York, 159-172, doi, 2012.

Ruppert, J., Mauder, M., Thomas, C., and Lüers, J.: Innovative gap-filling strategy for annual sums of $CO_2$ net ecosystem exchange, Agric. For. Meteorol., 138, 5-18, doi: 10.1016/j.agrformet.2006.03.003, 2006.

Schaller, C., Göckede, M., and Foken, T.: Flux calculation of short turbulent events – comparison of three methods, Atmos. Meas. Techn., 10, 869-880, doi: 10.5194/amt-10-869-2017, 2017.

Stull, R. B.: An Introduction to Boundary Layer Meteorology, Kluwer, Dordrecht, 666 pp., doi: 10.1007/978-94-009-3027-8, 1988.

Vickers, D., and Mahrt, L.: Quality control and flux sampling problems for tower and aircraft data, J. Atm. Oceanic Techn., 14, 512-526, doi: 10.1175/1520-0426(1997)014<0512:QCAFSP>2.0.CO;2, 1997.

Webb, E. K., Pearman, G. I., and Leuning, R.: Correction of the flux measurements for density effects due to heat and water vapour transfer, Quart. J. Roy. Meteorol. Soc., 106, 85-100, doi: 10.1002/qj.49710644707, 1980.

Wutzler, T., Lucas-Moffat, A., Migliavacca, M., Knauer, J., Sickel, K., Šigut, L., Menzer, O., and Reichstein, M.: Basic and extensible post-processing of eddy covariance flux data with REddyProc, Biogeosci., 15, 5015-5030, doi: 10.5194/bg-15-5015-2018, 2018.

Wyngaard, J. C., Coté, O. R., and Izumi, Y.: Local free convection, similarity and the budgets of shear stree and heat flux, J. Atmos. Sci., 28, 1171-1182, doi, 1971.

Yang, K., and Wang, J.: A temperature prediction-correction method for estimating surface soil heat flux from soil temperature and moisture data, Sci. China Ser. D: Earth Sci., 51, 721-729, doi, 2008.

---

## Author Response (AR2)

Dear Dr. Thonicke,

thank you for the final technical remarks and the handling of the manuscript. We have realized all desired corrections.

Best regards,

Thomas Foken